**COMMUNICATIONS**

# TGF-β signaling in Th17 cells promotes IL-22 production and colitis-associated colon cancer

Laura Garcia Perez[1], Jan Kempski[1], Heather M. McGee [2], Penelope Pelzcar[1], Theodora Agalioti[3], Anastasios Giannou[1], Leonie Konczalla [3], Leonie Brockmann[4], Ramez Wahib [3], Hao Xu[5], Maria Carolina Amezcua Vesely[5], Shiwa Soukou[1], Babett Steglich[1,3], Tanja Bedke[1], Carolin Manthey[1], Oliver Seiz[1], Björn-Philipp Diercks [6], Stylianos Gnafakis[7,8,9], Andreas H. Guse [6], Daniel Perez[3], Jakob R. Izbicki[3], Nicola Gagliani[1,3,10], Richard A. Flavell [5,11,12✉] & Samuel Huber [1,12✉]

IL-22 has dual functions during tumorigenesis. Short term IL-22 production protects against genotoxic stress, whereas uncontrolled IL-22 activity promotes tumor growth; therefore, tight regulation of IL-22 is essential. TGF-β1 promotes the differentiation of Th17 cells, which are known to be a major source of IL-22, but the effect of TGF-β signaling on the production of IL-22 in CD4+ T cells is controversial. Here we show an increased presence of IL-17+IL-22+ cells and TGF-β1 in colorectal cancer compared to normal adjacent tissue, whereas the frequency of IL-22 single producing cells is not changed. Accordingly, TGF-β signaling in CD4 + T cells (specifically Th17 cells) promotes the emergence of IL-22-producing Th17 cells and thereby tumorigenesis in mice. IL-22 single producing T cells, however, are not dependent on TGF-β signaling. We show that TGF-β, via AhR induction, and PI3K signaling promotes IL-22 production in Th17 cells.

[1] I. Department of Medicine, University Medical Center Hamburg-Eppendorf, 20246 Hamburg, Germany. [2] Center for Immunobiology and Microbial Pathogenesis, Salk Institute for Biological Studies, 10010 La Jolla, CA, USA. [3] Department of General, Visceral and Thoracic Surgery, University Medical Center Hamburg-Eppendorf, 20246 Hamburg, Germany. [4] Department of Microbiology and Immunology, Columbia University, New York, NY, USA. [5] Department of Immunobiology, School of Medicine, Yale University, New Haven, CT 06520, USA. [6] Department of Biochemistry and Molecular Cell Biology, University Medical Center Hamburg-Eppendorf, 20246 Hamburg, Germany. [7] Laboratory of Innate Immunity, Department of Microbiology, Infectious Diseases and Immunology, Charité - Universitätsmedizin Berlin, Hindenburgdamm 30, 12203 Berlin, Germany. [8] Berlin Institute of Health (BIH), Anna-Louisa-Karsch Strasse 2, 10117 Berlin, Germany. [9] Mucosal and Developmental Immunology, Deutsches Rheuma-Forschungszentrum, Charitéplatz 1, 10117 Berlin, Germany. [10] Immunology and Allergy Unit, Department of Medicine, Solna, Karolinska Institute and University Hospital, Stockholm, Sweden. [11] Howard Hughes Medical Institute, School of Medicine, Yale University, New Haven, CT 06520, USA. [12] These authors contributed equally: Richard A. Flavell, Samuel Huber. ✉email: richard.flavell@yale.edu; shuber@uke.de

nterleukin 22 (IL-22) has an important function in tissue repair and defense against pathogens. This cytokine is produced by immune cells and the selective expression of the receptor IL-22RA1 in non-immune cells restricts IL-22 functions to tissues specifically[1]. It contributes to the control of specific pathogens, such as *Citrobacter rodentium*[2,3]. Furthermore, IL-22 directly enhances the proliferation and survival of epithelial cells[4,5], which contributes to mucosal protection upon tissue damage. Interestingly, early IL-22 production protects against genotoxic stress[6]. However, if IL-22 is not controlled properly, it can also promote inflammatory bowel disease (IBD)[7] and tumorigenesis[8,9] in mouse models. Accordingly, elevated IL-22 serum levels have been associated with chemotherapy resistance in patients with colorectal cancer (CRC)[10]. IL-22 can be produced by several hematopoietic cell types. From the innate compartment the major sources of IL-22 are gamma delta T cells[11], natural killer cells[12] and innate lymphoid cells type 3 (ILC3)[9]. Within the adaptive immune system, Th17 cells were described to be an important source of IL-22[13,14]. Moreover, Th22 cells were identified as a distinct T-cell subset characterized by the production of IL-22 and the absence of IL-17 in humans[15,16]. Interestingly, T-cell derived IL-22 may be particularly important in CRC. First, the presence of IL-22-producing Th17 and Th22 cells has been shown to increase in later cancer stages in CRC patients, and to be correlated with a poor prognosis[17,18]. Second, it was reported that T-cell derived IL-22 can regulate the tumor niche by promoting the expression of core stem cell genes[19]; however, which T-cell subset contributed to cancer stemness was not addressed. By contrast, ILC3-derived IL-22 may be protective during early genotoxic events and thus against tumorigenesis[6]. Thus, the source of IL-22 may have a critical role for the function of IL-22.

Transforming growth factor beta1 (TGF-β1) has major effects on the differentiation of several T-cell subsets: it promotes the differentiation of peripheral regulatory T cells (pT$_{Regs}$) and Th17 cell[20–22], whereas it inhibits Th1 cell differentiation[23]. Moreover, TGF-β1 induces the expression of the transcription factor Aryl hydrocarbon receptor (AhR), a ligand-dependent transcription factor required for IL-22 production[3,15,24–26]. As for the impact of TGF-β1 on IL-22 production by T cells, the data are contradictory. On the one hand, previous work[3,27–29] suggests that TGF-β1 might suppress IL-22 production and specifically the differentiation of Th22 cells in vitro. On the other hand, other publications suggest that TGF-β1 promotes T-cell-derived IL-22 in vitro[13,24,30,31]. But in vivo evidence supporting either of these data are missing. TGF-β signaling is mediated by SMAD proteins. Upon binding to the TGF-βRII, TGF-β induces the phosphorylation of SMAD2 and/or SMAD3, which associate with SMAD4 that allows the translocation of this complex into the nucleus where it exerts the transcription of target genes[32]. Furthermore, non-canonical TGF-β signaling can interact with other signaling pathways. Most notably, TCR signaling and TGF-β signaling converge in the AKT/PI3K (protein kinase B/phosphoinositide-3 kinase)-mediated signaling cascade[33].

When T-cell receptor (TCR) recognizes its cognate antigen, the intracellular domains of the CD3 associated complex serve as site of interaction with other proteins that mediate the activation of PI3K, among many others[34]. Activation of PI3K leads to phospholipase Cγ-1 mediated generation of diacylglycerol (DAG) and inositol 1,4,5-trip-phosphate (IP3). Accumulation of intracellular IP3 leads to the opening calcium channels in the membrane originating a signaling cascade that activates transcription factors and modulates gene expression[35,36]. In this regard, it was demonstrated that not only do cytokines drive the polarization towards Th1 and Th2, but also the balance of Protein Kinase C (PKC) and calcium signaling driven by different TCR ligation is able to modulate this polarization[37,38]. Moreover, TCR signaling

is further enhanced by the second signal provided by the co-stimulatory molecule CD28. CD28 signaling uses a small subset of proteins already implicated in the TCR-mediated phosphorylation of substrates therefore amplifying the signal of the TCR, by for instance increasing calcium levels[36,39]. Taken together, several stimuli such as TGF-β signaling, TCR engagement or CD28 co-stimulatory signals can activate PI3K that in turn leads to an increased in intracellular calcium. However, the effect of these factors on the modulation of IL-22 production in T cells remains unknown.

Here we report that TGF-β1 and strong TCR stimulation, coupled with AhR ligands, promotes the emergence of IL-17A+IL-22+ T cells and the production of IL-22 in already differentiated Th17 cells in vitro. Moreover, using transgenic mice with impaired TGF-β signaling in T cells and IL-17A+ producing cells, we demonstrate that TGF-β signaling promotes IL-22 production by Th17 cells and thus intestinal tumorigenesis in vivo.

## Results

**IL-17A+IL-22+ T cells are enriched in CRC.** TGF-β1 is an important cytokine within the tumor microenvironment and high levels correlate with poor prognosis in CRC patients[40]. In order to corroborate this finding, we measured the concentration of total and active TGF-β1 in tissue lysates from colon tumor and normal adjacent tissue from a cohort of patients with sporadic CRC (Supplementary Table 1). Interestingly, we found that the total amount of TGF-β1, and the active form of TGF-β1 was increased in the tumor tissue, compared with normal adjacent tissue control (Fig. 1A). We then analyzed the presence of IL-22 producing CD4+ T cells isolated from the same cohort of patients. The analysis of fresh CRC specimens (Supplementary Table 1) indicated an increased infiltration of CD4+ T cells producing IL-22 by trend compared with adjacent normal colon tissue (frequency of IL-22-producing cells within CD4+ T cells: control: 5.09±5.5 vs tumor: 7.68±6.21, $p = 0.0879$). Further characterization of these cells showed that the increased IL-22 frequency in the tumor was owing to an enrichment of IL-17A+IL-22+ double producing CD4+ T cells, whereas IL-17A-L-22+ single producing T cells were not changed (Fig. 1B–D). Taken together, both IL-17A+IL-22+ producing T cells as well as TGF-β1 levels are increased in human CRC samples compared to normal adjacent tissue. Based on these data, we hypothesized that TGF-β1 might promote the emergence of IL-17A+IL-22+CD4+ T cells in CRC.

**TGF-β1 promotes IL-17A+IL-22+ CD4+ T cells in vitro.** We next aimed to test the role of TGF-β1 for the production of IL-22 in vitro. To that end, we differentiated naive T cells, purified with an efficiency around 85% using magnetic cell separation (MACS) beads, from wild-type mice (WT) under different conditions, that have been previously reported to modulate IL-22 production in vitro[27–29,31,41]. In particular, we were interested in the effect of TGF-β1, 6-formylindolo[3,2-b]carbazole (FICZ, an AhR ligand) and IL-1β. In addition, we tested the impact of different strengths of TCR stimulation, as this also impacts T-cell differentiation[42]. To this end, we cultured naive T cells in the presence of antigen-presenting cells (APCs), anti-CD3, and IL-6. TGF-β1, FICZ, and IL-1β were added as indicated (Fig. 2). In line with the human data, we found that the addition of TGF-β1 increased the expression of IL-22 and IL-17A (Fig. 2A, B, Supplementary Fig. 1). As for IL-22, this effect was, however, dependent on the addition of anti-CD28 (Fig. 2A, B; Supplementary Fig. 1). Adding FICZ resulted in a further increase in IL-22 expression especially in the presence of TGF-β1 (Fig. 2A, B). Adding IL-1β did increase IL-22 levels in the presence of low TCR stimulation, but not in the presence of high TCR stimulation. Thus, TGF-β1 promoted

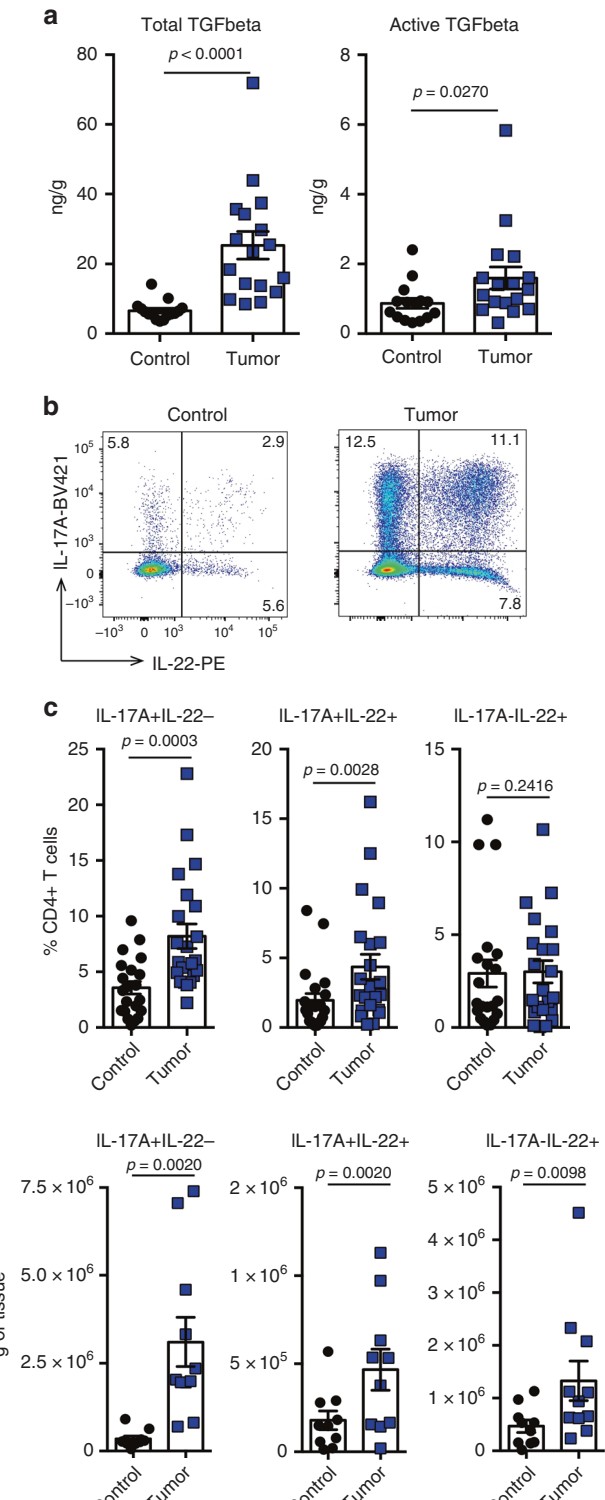

**Fig. 1 IL-17+IL-22+ T cells are enriched in human colorectal cancer samples. a** Protein level of total and active TGF-β1 was measured by ELISA in lysates from frozen tissue of colorectal cancer samples and normal adjacent tissue (control). Each dot represents one patient (control $n = 15$; tumor $n = 17$). Bars represent mean, error bars show±sem. **b–d** Cells were isolated from fresh tissue of colorectal cancer samples and normal adjacent tissue (control) and analyzed by flow cytometry. **b** Representative FACS plots of IL-17A and IL-22 expression. **c** Frequencies (control $n = 21$; tumor $n = 22$) and **d** cell number (control $n = 10$; tumor $n = 11$) of IL-17+IL-22-, IL-17+IL-22+, and IL-17-IL-22+ within CD4+ T cells is shown. Bars represent mean, error bars show±sem. **a**, **c**, **d** Two-sided Mann–Whitney test was performed to assess significance ($P < 0.05$). Source data are provided as a Source data file.

of IL-17A+IL-22+ double producing T cells (Fig. 2C, D). These IL-17A+IL-22+ cells were dependent on the presence of TGF-β1 and were further promoted by FICZ, reaching the highest level in the presence of stronger TCR stimulation (Fig. 2C, D). The addition of IL-1β slightly increased the frequency of this population in the absence of strong stimulation but it did not have an additive effect on IL-17A+IL-22+ cells in the presence of a stronger stimulation (Fig. 2C, D). The optimal TGF-β1 concentration for IL-22 induction was further evaluated in a titration experiment (Supplementary Fig. 3). Moreover, the induction of Foxp3+ inducible regulatory T cells (iT_{Reg}) in this setting was minimal (Supplementary Fig. 3). Addition of IL-23 did not potentiate the effect of IL-1β and FICZ in the induction of IL-22 (Supplementary Fig. 3).

Overall, we found that TGF-β1 and FICZ, in the presence of IL-6 and strong stimulation, promoted the emergence of IL-17A+IL-22+CD4+ T cells in vitro.

**TGF-β promotes IL-17A+IL-22+ T cells in vivo.** In order to assess the role of TGF-β1 in the regulation of IL-22 production by CD4+ T cells in vivo, we used a transgenic mouse model over-expressing a dominant negative TGF-βRII in T cells (TGF-β-DNR). In this mouse model all CD4+ and CD8+ T cells have impaired TGF-β signaling[21]. We furthermore used a colitis-associated colon cancer mouse model in which the administration of the mutagen AOM (azoxymethane) and the induction of a chronic colitis using dextran sulfate sodium (DSS) promotes tumorigenesis in the intestine[8]. In order to discriminate between cell extrinsic and cell intrinsic effects, we co-transferred congenic wild-type CD4+ T cells, either with congenic wild-type or TGF-β-DNR transgenic (Tg) CD4+ T cells into Rag1−/− mice (Fig. 3A). Upon reconstitution we induced colitis-associated colon cancer. We found that the frequency of IL-17A+IL-22− and IL-17A+IL-22+ CD4+ T cells was significantly reduced in TGF-β-DNR transgenic CD4+ T cells compared to co-transferred wild-type cells in the same environment (gating strategy shown in Supplementary Fig. 8). As expected[43], the frequency of Foxp3+ CD4+ T cells (Supplementary Fig. 4) was also reduced in TGF-β-DNR transgenic CD4+ T cells compared with wild-type control in normal colon. However, this was not the case in the tumor tissue. Interestingly, the presence of IL-17A+Foxp3+ T cells in the tumors was not affected by the impaired TGF-β signaling. In contrast, the frequency of IL-17A-IL-22+CD4+ T cells was unaffected by the impairment of TGF-β signaling (Fig. 3A). Of note, these results were not restricted to colon cancer but were also confirmed in C. rodentium colitis using a similar approach (Supplementary Fig. 5).

Next, we analyzed whether T-cell specific impairment in TGF-β-signaling influences tumorigenesis. In order to exclude the effect of non-T-cell derived IL-22, which has been recently

IL-22 production in vitro in the presence of IL-6 and a strong stimulation in mouse T cells in vitro.

In order to validate these results and to evaluate the production of IL-22 by different T-cell subsets, we used the same in vitro setting using Foxp3^{mRFP} × IL-17A^{eGFP} × IL-22^{sgBFP} triple reporter mice (Supplementary Fig. 2). TGF-β1 was important for the induction of IL-17A. Also, in line with previous publications[3,28], we found that TGF-β1 reduced the emergence of IL-22 single producing T cells at least in the presence of a low TCR stimulation. However, TGF-β1 promoted the differentiation

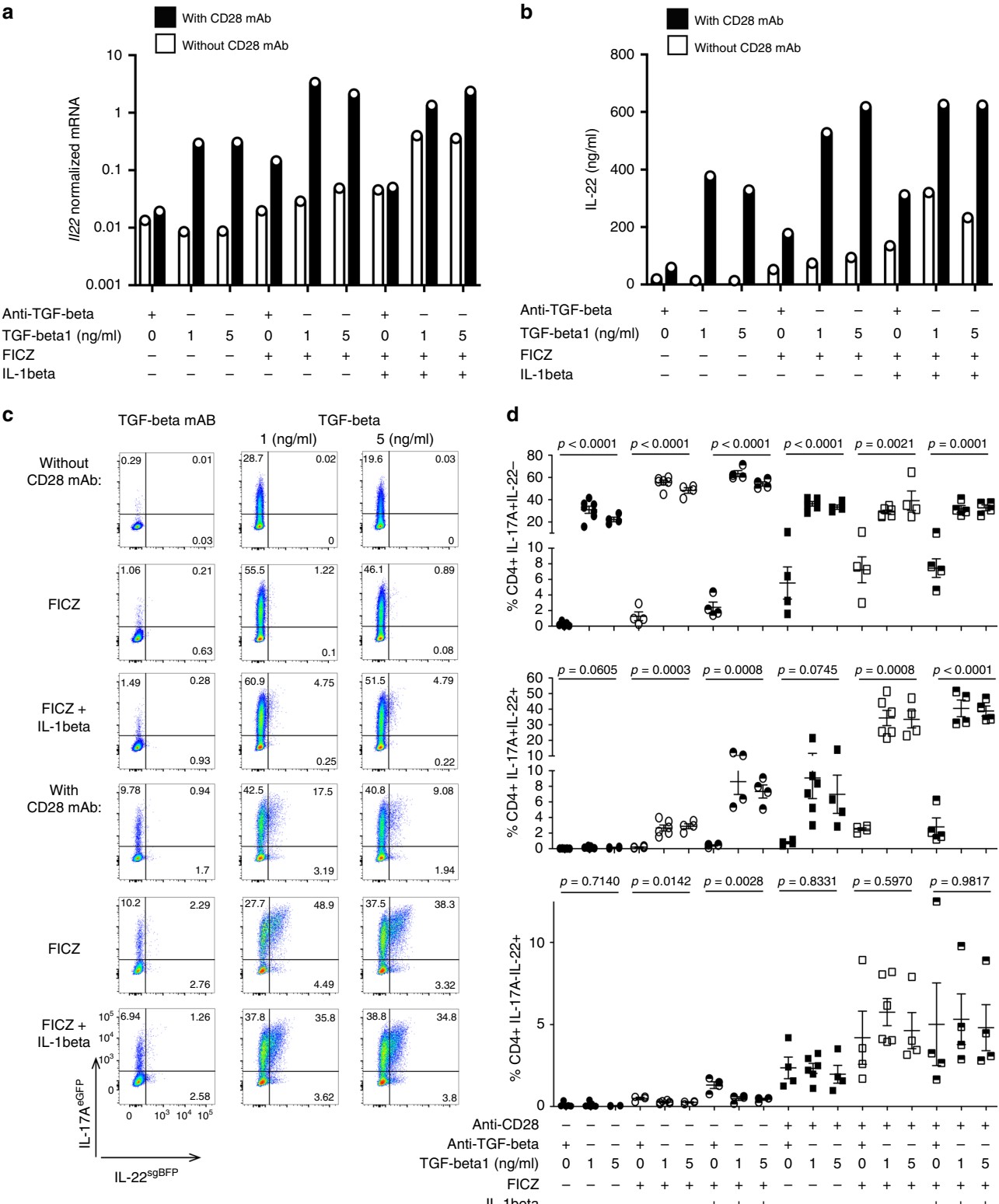

**Fig. 2 TGF-β1 promotes IL-17+IL-22+ producing T cells in vitro.** Naive T cells were isolated from spleen and lymph nodes of wild-type (C57BL/6 J) mice and cultured for 4 days under indicated conditions. Relative *Il22* mRNA **a** and IL-22 protein level as measured by ELISA in cell culture supernatants **b**, mean of technical duplicates is shown, representative of two independent experiments. Naive T cells were isolated from spleen and lymph nodes of Foxp3mRFP × IL-17AeGFP × IL-22sgBFP reporter mice and cultured for 4 days under indicated conditions. Representative FACS plots **c** and statistics **d** of indicated cell populations as assessed by flow cytometry. Each dot represents the result from one experiment. n = 4–7 dependent on the respective condition. The results are cumulative from four to seven independent experiments depended on the respective condition. Bars represent mean±sem. One-way ANOVA was performed to assess significance (P < 0.05). Source data are provided as a Source data file.

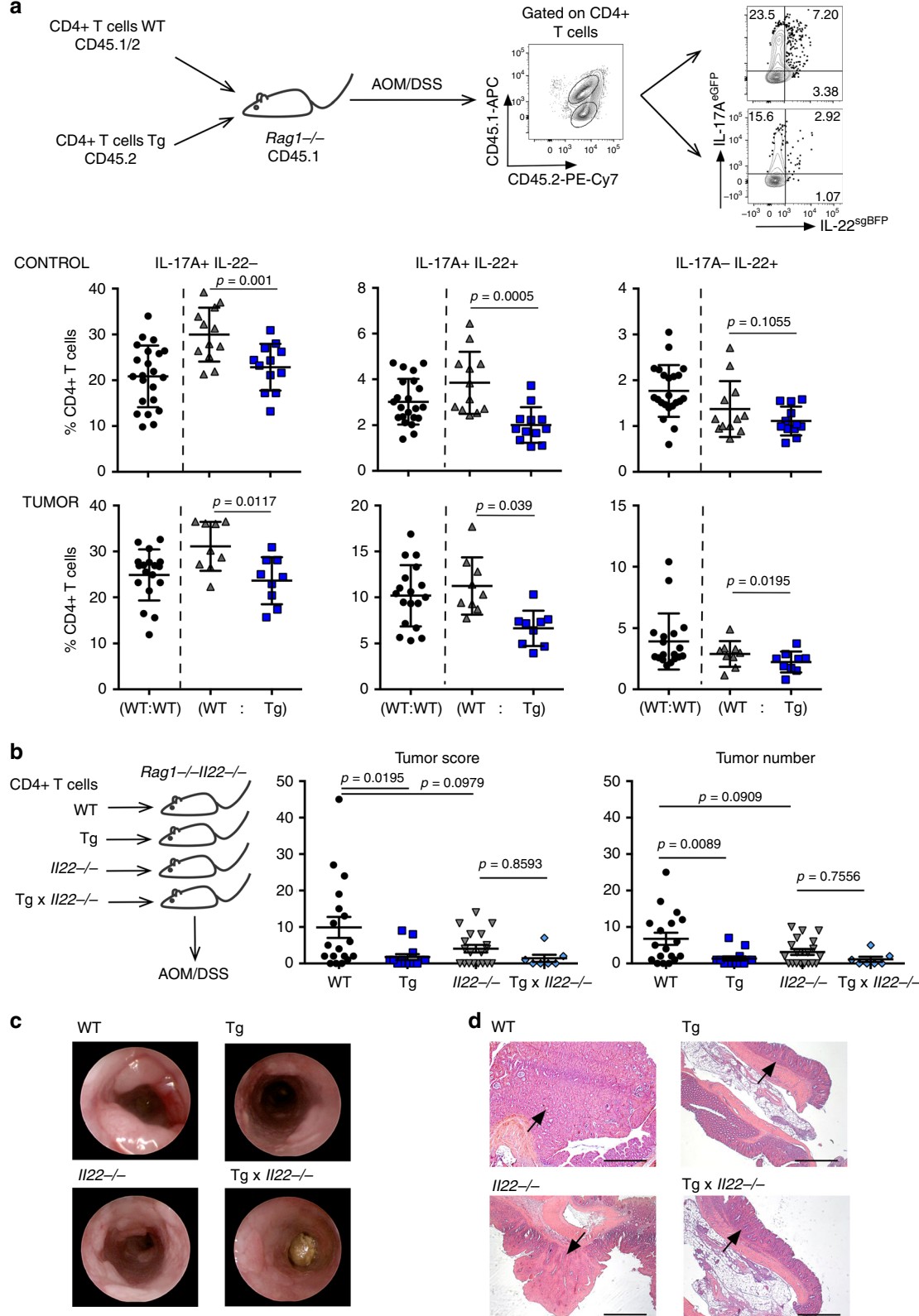

reported to have a protective function in CRC[6] and also to exclude effects of TGF-β on CD8+ T cells, we repopulated *Il22*-deficient lymphopenic hosts (*Rag1*−/−*Il22*−/−) with either wild type (WT), TGF-β-DNR transgenic (Tg), *Il22*−/− or TGF-β-DNR transgenic × *Il22*−/− (Tg × *Il22*−/−) CD4+ T cells and induced colitis-associated colon cancer upon engraftment using the AOM/DSS model. Mice receiving wild-type T cells showed a higher tumor load compared with mice receiving TGF-β-DNR transgenic CD4+ T cells (Fig. 3). Moreover, in an IL-22-free environment mice receiving *Il22*−/− or Tg × *Il22*−/− CD4+ T cell showed an equal tumor load (Fig. 3), indicating that the observed effect is IL-22 dependent.

In conclusion, TGF-β signaling in CD4+ T cells promotes the emergence of IL-17A+IL-22+ T cells in a direct manner in vivo.

**Fig. 3 TGF-β signaling promotes the emergence of IL-17+IL-22+ T cells in a direct manner in vivo. a** Congenic CD4+ T cells from Foxp3[mRFP] × IL-17A[eGFP] × IL-22[sgBFP] or Foxp3[mRFP] × IL-17A[eGFP] × IL-22[sgBFP] × dnTGF-βR2 (Tg) mice were co-transferred into *Rag1−/−* prior to tumor induction using AOM/DSS. Production of IL-17A and IL-22 by T cells was analyzed in tumors and normal adjacent tissue (control) using flow cytometry. Results are cumulative from two independent experiments. Control: (WT:WT) $n = 22$; (WT:Tg) $n = 12$. Tumor: (WT:WT) $n = 18$; (WT:Tg)=9. Lines indicate mean ±sem. Two-sided Wilcoxon multiple comparisons test was performed ($P < 0.05$) to assess the significance. **b** Colitis-associated colon cancer was induced in *Rag1−/−* × *Il-22−/−* mice upon reconstitution with wild type (WT), dnTGF-βR2 (Tg), *Il-22−/−* or dnTGF-βR2 × *Il-22−/−* (Tg × *Il22−/−*) CD4+ T cells. Tumor score and tumor number is shown. **c** Representative endoscopic view and **d** histological sections are shown, representative from two independent experiments; arrows indicate tumors; scale bar represents 500 μm. Results are cumulative from two independent experiments. Each dot represents one mouse (WT $n = 18$; Tg $n = 14$; *Il22−/−* $n = 19$; Tg × *Il22−/−* $n = 7$). Lines indicate mean±sem; One-way ANOVA, Tukey´s multiple comparisons test was performed ($P < 0.05$) to assess the significance. Source data are provided as a Source data file.

Furthermore, this correlates with an increased tumorigenesis in vivo.

**TGF-β signaling in Th17 cells promotes IL-22 production**. One limitation of the aforementioned experiments was that all CD4+ T cells have an impaired TGF-β signaling. Thus, it is not possible to discriminate between the effect of TGF-β on naive T cells and already differentiated Th17 cells. To overcome this boundary, we next used IL-17A[Cre] × TGFBR2[fl/fl] mice in which TGF-β signaling is ablated in cells that express IL-17A[44]. In order to discriminate between cell intrinsic and cell extrinsic effects and also to restrict the deletion of the TGF-βRII to IL-17A-producing CD4+ T cells, we co-transferred wild-type CD4+ T cells with congenic wild-type or IL-17A[Cre] × TGFBR2[fl/fl] CD4+ T cells into *Rag1−/−* mice (Fig. 4A). Upon reconstitution, we induced colitis-associated colon cancer using AOM/DSS. In line with our results using TGF-β-DNR transgenic CD4+ T cells, we found a reduced frequency of IL-17A+IL-22− and IL-17A+IL-22+ T cells in the transgenic compared with the wild-type T-cell fraction (Fig. 4A). Interestingly, the frequency of IL-17A-IL-22+ producing T cells was increased in CD4+ T cells with blocked TGF-β signaling (Fig. 4A), suggesting that Th17 cells might, in principle, be able to convert into IL-22 single producing cells. To test this hypothesis, we crossed Fate+ mice[44] with IL-22[sgBFP] reporter mice (IL-17A[Cre] × Rosa26[YFP] × IL-17A[FP635] × IL-22[sgBFP]). Interestingly, we found that some IL-17-IL-22+ cells were yellow fluorescence protein (YFP)+, indicating that Th17 cells are in principle able to downregulate IL-17 production while maintaining IL-22 production. However, the vast majority of IL-22+ single producing cells in tumors were YFP−, indicating that these cells do not come from IL-17A+ cells (Supplementary Fig. 6). Finally, we wanted to evaluate whether the blockade of TGF-β-signaling in IL-17A+ CD4+ T cells would also impact tumor development. Therefore, we induced colitis-associated colon cancer in IL-17A[Cre] × TGFBR2[fl/fl] and control mice. Interestingly, we found that IL-17A[Cre] × TGFBR2[fl/fl] had a lower tumor score compared with their wild-type littermate controls (Fig. 4B–D).

In conclusion, TGF-β-signaling in Th17 cells promotes IL-17A+IL-22− and IL-17A+IL-22+ T cells, and tumorigenesis during colitis-associated colon cancer in a mouse model.

**TGF-β induces the expression of AhR in Th17 cells**. Based on the above-mentioned results, we next wished to identify the molecular mechanism by which TGF-β promotes IL-17A+IL-22+ CD4+ T cells. To this end, we used a supervised approach and evaluated the expression of genes that had been linked to IL-17A and IL-22 production in T cells before[3,29]. As control, we also analyzed IL-10, which as IL-22, belongs to the IL-10 cytokine family. To this end, we isolated naive T cells from WT mice and cultured them in vitro under the indicated conditions (Fig. 5A). In line with the results shown in Fig. 2, we confirmed that *Il22*

mRNA expression was highest in T cells, which had been cultured in the presence of IL-6, TGF-β1, FICZ, and strong TCR stimulation, whereas strong TCR stimulation was not crucial for *Il17a* expression. In contrast, *Il10* mRNA was higher in the presence of IL-6, TGF-β1, FICZ, and weaker TCR stimulation, whereas *Il17a* mRNA was equally expressed in both conditions. Interestingly, high *Il22* mRNA correlated with high *Ahr* and RAR-related orphan receptor c (*Rorc*) expression, but lower avian musculoaponeurotic fibrosarcoma oncogene homolog (*cmaf*) expression (Fig. 5A). In contrast, *Il10* mRNA correlated with high *Ahr* and *cmaf* expression (Fig. 5A).

To further evaluate the role of the transcription factor *Ahr* in the development of IL-17A and IL-22-producing T cells, we differentiated naive T cells under Th17-polarizing conditions in the presence of increasing amounts of an AhR inhibitor. The inhibition of AhR strongly reduced the frequency of IL-17+IL-22+ cells compared with control, whereas the frequency of IL-17+IL-22− and IL-17-IL-22+ was slightly but not significantly reduced (Fig. 5B). In order to confirm our results, we differentiated naive T cells from Rorgt[Cre] × AhR[fl/fl] × ROSA[YFP] mice[45], which have a deletion of AhR in all T cells[46]. The deletion of AhR in T cells significantly reduced the emergence of IL-17+IL-22− and IL-17+IL-22+ cells, whereas IL-17-IL-22+ were not affected (Fig. 5C). In order to discriminate between effects on naive T cells and already differentiated Th17 cells, we isolated naive T cells from Foxp3[mRFP] × IL-17A[eGFP] × IL-22[sgBFP] triple reporter mice and cultured them under Th17 cell-polarizing conditions. Upon in vitro differentiation we sorted Foxp3-IL-17A+IL-22− T cells and re-stimulated them in the presence TGF-β1 alone or with FICZ. As control, we added a neutralizing TGF-β antibody and an AhR inhibitor (Fig. 5D). We found that TGF-β1 was important for the maintenance of IL-17A expression (Fig. 5D). More importantly, we found that TGF-β1 and AhR signaling were essential for Th17 cells to acquire the production of IL-22 in vitro. The frequency of IL-17-IL-22+ cells was very low, but it was slightly increased in the presence of AhR ligand and the blocking TGF-β1 antibody (Fig. 5D), confirming previous results[29], indicating a negative effect of TGF-β1 on IL-17A-IL-22+ T cells in vitro.

These results indicate that TGF-β1 and AhR signaling promote the induction of IL-17+IL-22+ CD4+ T cells in vitro.

**PI3 kinase promotes IL-17+IL-22+ T-cell differentiation**. Finally, we aimed to understand why a strong TCR stimulation was critical for the efficient induction of IL-22-producing CD4+ T cells. Phosphoinositide-3 Kinase (PI3K) is an important downstream mediator of the TCR and CD28 co-stimulatory signal, therefore we aimed to investigate the role of this kinase in the regulation of IL-22 production in T cells. In order to assess the impact of the strength of TCR stimulation on IL-22 production, we first measured the intracellular calcium concentration of CD4+ T cells under different TCR stimulations. To this end, we performed fluorescence microscopy of Fura-2-labeled

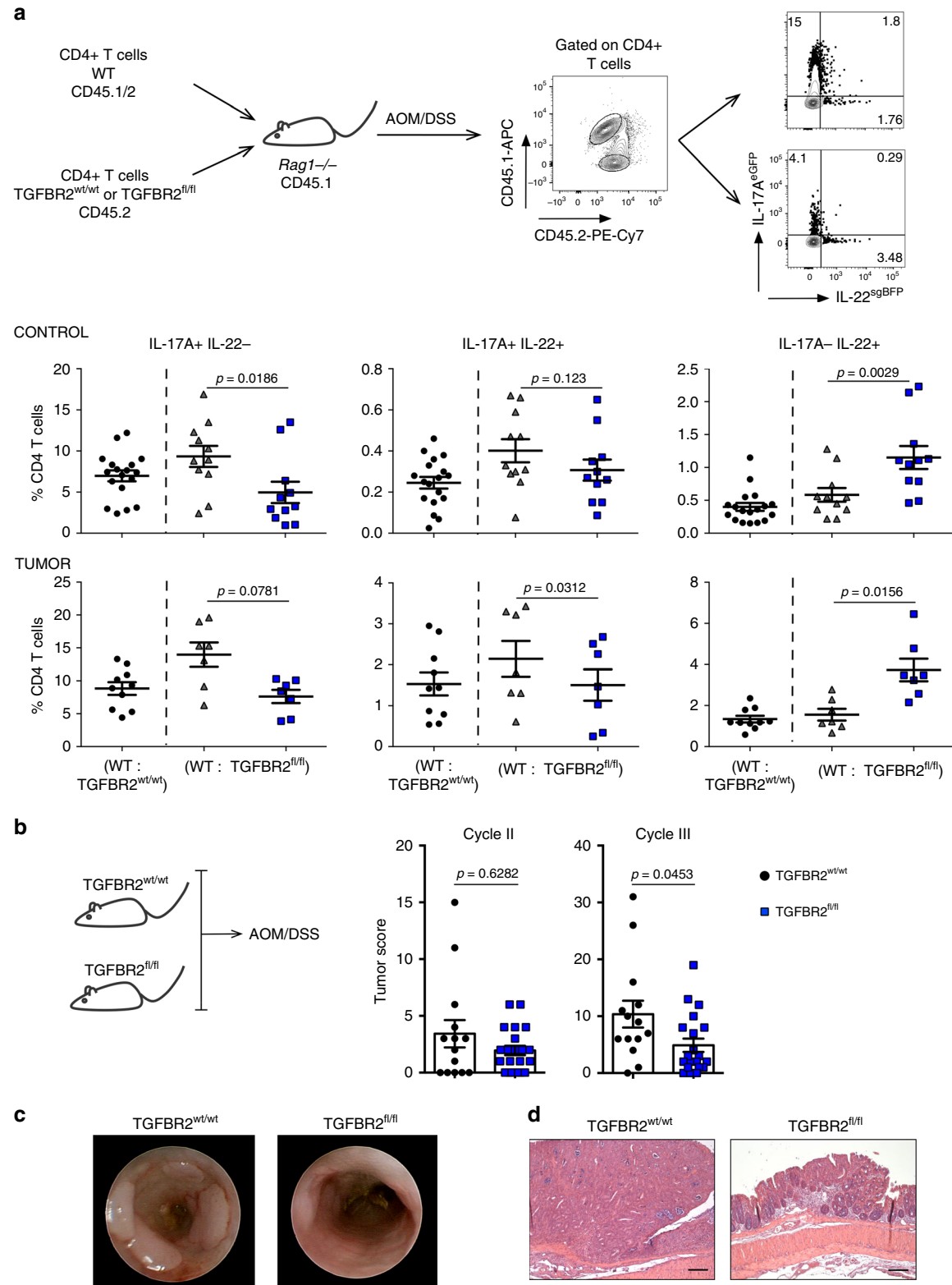

CD4+ T cells (a fluorescent dye that binds to free intracellular calcium) overlaid with unlabeled APCs (Fig. 6A). As expected, a strong stimulation corresponded with higher calcium accumulation compared with a weaker stimulation. To analyze the role of PI3K, we used a chemical inhibitor (Wortmannin (W)). Interestingly, the addition of the PI3K inhibitor significantly reduced the calcium accumulation mediated by the strong TCR stimulation (Fig. 6B, C) to the level of the weak TCR stimulation. Next,

to investigate the link between PI3K-mediated calcium accumulation and the production of IL-22, we differentiated naive T cells under Th17-polarizing conditions in the presence of increasing amounts of the PI3K inhibitor. In order to avoid the need to re-stimulate cells to detect cytokines, we used a Foxp3$^{mRFP}$ × IL-10$^{eGFP}$ × IL-17$^{Kat}$ × IL-22$^{sgBFP}$ reporter mouse model. Strikingly, we found that the inhibition of PI3K caused a reduced frequency of IL-17+IL-22+ cells, whereas the frequency

**Fig. 4 TGF-β signaling in Th17 cells promotes IL-22 production. a** CD4+ T cells from Foxp3$^{mRFP}$ × IL-17A$^{eGFP}$ × IL-22$^{sgBFP}$, IL-17A$^{eGFP}$ × IL-22$^{sgBFP}$ × IL-17A$^{Cre}$ × TGFBR2$^{fl/fl}$ or IL-17A$^{eGFP}$ × IL-22$^{sgBFP}$ × IL-17A$^{Cre}$ × TGFBR2$^{wt/wt}$ mice were co-transferred into *Rag1−/−* prior to tumor induction. Production of IL-17 and IL-22 by T cells was analyzed by flow cytometry in both normal adjacent tissue (control) and tumors. Results are cumulative from two independent experiments. Control: (WT: TGFBR2$^{wt/wt}$) $n = 18$; (WT:TGFBR2$^{fl/fl}$) $n = 11$. Tumor: (WT: TGFBR2$^{wt/wt}$) $n = 10$; (WT:TGFBR2$^{fl/fl}$) $n = 7$. Lines indicate mean±sem. Two-sided Wilcoxon multiple comparisons test was performed ($P < 0.05$) to assess the significance. **b** Colitis-associated colon cancer was induced in IL-17A$^{Cre}$ × TGFBR2$^{fl/fl}$ or IL-17A$^{Cre}$ × TGFBR2$^{Wt/Wt}$ mice. Tumor score after second and third DSS-cycle is shown. **c** Representative endoscopic view and **d** histological sections at the end of cycle III are shown, representative from two independent experiments; squares show tumors; scale bar represents 50 μm. Results are cumulative from two independent experiments. Each dot represents one mouse (TGFBR2$^{wt/wt}$ $n = 14$; TGFBR2$^{fl/fl}$ $n = 21$). Lines indicate mean± sem; Two-sided Mann–Whitney test was performed ($P < 0.05$) to assess significance. Source data are provided as a Source data file.

of IL-17+IL-22− cells were not significantly affected (Fig. 6D). The emergence of IL-17-IL-22+ T cells was by trend reduced by the inhibition of PI3K. These data are in line with the in vitro data showing that induction of IL-17A+IL-22+ and to a lesser extent IL-17A-IL-22+, but not IL-17A+IL-22− T cells depend on strong TCR stimulation (Fig. 2D). As control, we also measured the role of PI3K on the emergence on IL-10-producing T cells. In line with the data shown in Fig. 5A, we did not observe a significant effect of PI3K inhibition on the emergence of IL-10-producing T cells (Supplementary Fig. 7).

## Discussion

CRC, like most solid tumors, present with immune cell infiltrates that influence the outcome: infiltration of Th1 and CD8+ T cells correlates with a good prognosis; however, infiltration of Th17 cells has been suggested to promote tumorigenesis[47] and is associated with reduced disease-free survival in CRC patients[48]. Specifically, IL-17A mediates the progression of intestinal cancer in mouse models[49] acting on transformed enterocytes[50] and with IL-6 and tumor necrosis factor-alpha (TNF)-promoting CRC cell growth via signal transducer and activator of transcription 3 (STAT3) and nuclear factor-kappa-B (NFκB) activation[51]. IL-22 has dual effects during tumorigenesis in the intestine. IL-22 has protective effects during early genotoxic events[6]. However, IL-22 is also known to promote tumor growth[8,9,19]. These data highlight again the need to control IL-22 production.

Interestingly, Th17 cells have been described to produce IL-22 in homeostatic conditions and upon *C. rodentium* infection[52]. However, the specific function of these cells in CRC and the mechanisms regulating them remained elusive. Furthermore, it was unclear whether IL-17A and IL-22 are co-produced by the same cells or by different cells and unclear, which factors would regulate IL-22 expression in the respective cells. These points are critical when it comes to designing new therapies targeting IL-17A and IL-22. Indeed, we found an increased frequency of IL-17A and IL-22 producing CD4+ T cells in the tumor compared with adjacent normal tissue in patients with CRC. Interestingly, the largest increase was in IL-17A+IL-22− and IL-17A+IL-22+ double producing cells, whereas the frequency of IL-17A-IL-22+ single producing cells was not altered. These data are in line with a recent publication, which showed an increased frequency of Th17, which co-produced IL-17+IL-22+ in CRC patients[17]. Also, the higher frequency of IL-17A+IL-22+ producing CD4+ T cells correlated with later stages of the cancer[17]. However, despite these studies the factors that regulate their emergence were still unknown.

We focused on TGF-β1, as this cytokine is known to have broad functions within the immune system and particularly on CD4+ T cells[53]. Furthermore, TGF-β is present in the tumor microenvironment and correlates with poor prognosis in CRC patients[40]. However, the role of TGF-β for the emergence of IL-17A+IL-22-, IL-17A+IL-22+, and IL-17-IL-22+ CD4+ T cells in the tumor microenvironment remained unknown and was

addressed by our study. We confirmed that both active and total TGF-β1 is increased in tumor tissue in patients with CRC compared with normal adjacent tissue. More importantly, we found that TGF-β signaling in T cells promotes the emergence of IL-17A+IL-22-, IL-17A+IL-22+, but not IL-17-IL-22+ CD4+ T cells in vitro and in vivo and this correlated with an increased tumorigenesis. Of note, TGF-β can also act on other immune cells besides CD4+ T cells and several non-immune cells, thereby impacting carcinogenesis, however, this was not the focus of this study.

In humans, Th22 cells were described to be differentiated from their precursors by the combination of IL-6 and TNF-alpha. Vitamin D further enhances the production of IL-22[15,16]. Another report showed that Th17 cells generated in the absence of TGF-β1 also produce IL-22[54]. In line with this, it was shown that TGF-β1 induces IL-17A and reduces IL-22, respectively, in a dose-dependent manner[14]. In contrast, it was described later on that the combination of TGF-β1 and AhR ligands induces the production of IL-22 from lamina propria lymphocytes[25]. In the murine system, several cytokines have been proposed to induce IL-22 in vitro. First, it was described that the combination of IL-6, IL-23, and IL-1β leads to the generation of Th17 cells and IL-22[27]. Moreover, Rutz et al.[29] showed that the production of IL-22 in Th17 cells is inhibited by TGF-β1 and the combination of IL-6, IL-23 and neutralizing TGF-β1 antibody was required to obtain IL-22 single and IL-17A+IL-22+ co-producing CD4+ T cells in vitro[29]. In line with these data, Basu et al.[3] reported that optimal IL-22 production in Th22 cells is acquired in the absence of TGF-β1. Accordingly, the addition of a TGF-βR2 inhibitor to a cocktail containing IL-6, IL-23, IL-1β, and AhR ligands, strongly induces IL-22 single producing cells in vitro[28]. In contrast, other reports have shown the production of IL-22 in Th17 cells using cocktails containing TGF-β1[30,41,55]. Taken together, the role of TGF-β1 on the production of IL-22 in vitro is contradictory. And our aim was to solve these contradictions. Thus, we studied the role of TGF-β1 using reporter mice for IL-17A and IL-22 and in vitro differentiation experiments that allowed us to reliably monitor the expression of IL-17A and IL-22 without the need for in vitro re-stimulation. Overall, we found that TGF-β1 has a different impact on the emergence of IL-22 producing Th17 and Th22 cells. It induces the differentiation of IL-17+IL-22− and IL-17A+IL-22+ CD4+ T cells, whereas IL-17-IL-22+ single producing CD4+ T cells were not dependent on TGF-β signaling. We furthermore deciphered the mechanisms, which might explain some of the controversial results mentioned above. Based on our results, not only the cytokine milieu, but also the strength of TCR engagement is important for the emergence of IL-17A+IL-22+ CD4+ T cells. In line with this, the promoting effect of TGF-β1 on the generation of IL-17A+IL-22+ CD4+ T cells was obtained in vitro only in the presence of strong TCR stimulation. Strong TCR signaling correlated with higher and longer accumulation of intracellular calcium during the early activation state, mediated by PI3K. Interestingly, activation of PI3K and calcium

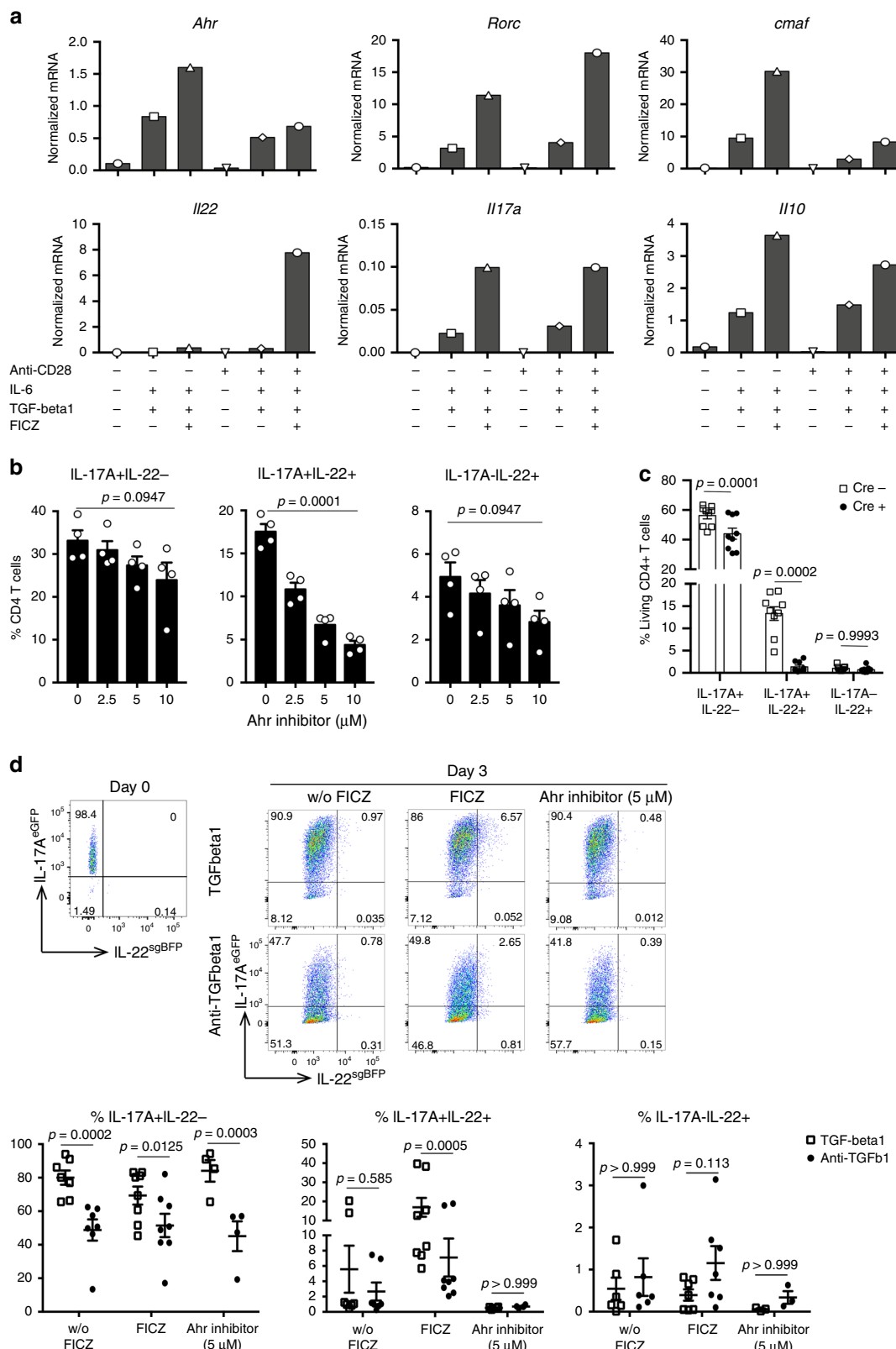

(Ca2+) influx was seen in the absence of TGF-β. In the performed experiments, we used a calibration buffer to resuspend the cells, which is serum free and therefore also TGF-β free. Thus, TGF-β signaling is not essential for PI3K activation. However, TGF-β signaling is important for *Ahr* induction, which is then

essential to induce IL-22 production (Fig. 5). Thus, TGF-β signaling in conjunction with strong TCR stimulation was important to promote the transcription of IL-22.

Taken together, our results suggest that although IL-17A+IL-22− cells and IL-17A+IL-22+ cells depend on TGF-β, they have

**Fig. 5 AhR signaling mediated the effects of TGF-β1 on the emergence of IL-17+IL-22+ T cells. a** Naive T cells were differentiated in the presence of anti-CD3 (3 μg/ml), APCs and indicated factors. Relative *Ahr, Rorc, cmaf, Il22, Il17a, Il10* mRNA expression on day 2 of the culture was measured using RT-PCR, mean of technical duplicates is shown, results are representative of two independent experiments. **b** Naive T cells from Foxp3^mRFP × IL-17A^eGFP × IL-22^sgBFP reporter mice were cultured under Th17-polarizing condition (mAb IL-4 (10 μg/ml), mAb INF-γ (10 μg/ml), mAb CD3 (3 μg/ml), mAb CD28 (0.5 μg/ml), IL-6 (10 ng/ml), TGF-β1 (1 ng/ml), and FICZ (100 mM)) with increasing amounts of AhR antagonist (as indicated) for 4 days. Frequency of indicated cell populations are shown. Bars represent mean, error bars show±sem, *n* = 4. One-way ANOVA, Dunnett's multiple comparisons test. Data are cumulative from three independent experiments. **c** Naive T cells from Rorgt^Cre × AhR^fl/fl × ROSA^YFP and control mice were cultured under Th17-polarizing condition for 4 days. Frequency of indicated cell populations are shown. Bars represent mean, error bars show±sem, *n* = 9. Two-way ANOVA, Sidak's multiple comparisons test. Data are cumulative from three independent experiments. **d** In vitro differentiated Th17 cells (IL-17A+IL-22-) from Foxp3^mRFP × IL-17A^eGFP × IL-22^sgBFP reporter mice were sorted and re-cultured with a standard stimulation (mAb CD3 (3 μg/ml), mAb CD28 (0.5 μg/ml), IL-6 (10 ng/ml)) in the presence or absence of TGF-β and Ahr ligand (FICZ) or antagonist. Representative FACS plots upon re-culture in indicated conditions (left panel) and statistics (right panel) are shown, w/o FICZ *n* = 7, FICZ = *n* = 8 and Ahr inhibitor *n* = 4. Two-way ANOVA, Bonferroni's multiple comparisons test. Bars represent mean, error bars show±sem. Data are cumulative from four independent experiments. Source data are provided as a Source data file.

different additional requirements for their development. IL-17A +IL-22+ cells are dependent on AhR activation and strong TCR signaling mediated PI3K activation.

The effect of TGF-β signaling on IL-22 production by T cells might be different in vitro and in vivo. Indeed, we did observe a high variability in the frequency of IL-17+IL-22+ CD4+ T cells over the course of the in vivo experiments. This effect might be owing to different intestinal microbiota compositions of the mouse lines used. We therefore used littermate controls and co-transfer experiments in order to control for microbial effects. Thus, using a transgenic mouse model overexpressing a dominant negative TGF-βRII in T cells (TGF-β-DNR), we again found that TGF-β signaling in T cells promoted the emergence of IL-17+IL-22- and IL-17A+IL-22+ CD4+ T cells, whereas IL-17A-IL-22+ single producing CD4+ T cells were not affected. The reduced frequency of IL-17A+IL-22- cells in transgenic mice corroborates the important role of TGF-β for the development and stability of Th17 cells. Despite the fact that Th17 cells can be generated in the absence of TGF-β[27], our in vivo and in vitro results support the role of TGF-β for stabilizing IL-17A production by Th17 cells. Furthermore, one could assume that the reduced amount of IL-17A+IL-22+ co-producing CD4+ T cells in the absence of TGF-β signaling could be due to the fact that Th17 cells per se are affected and they have a reduced frequency and consequently, also IL-22 producing Th17 are reduced. To exclude this, we used a model with Th17 cell specific ablation of TGF-β signaling, confirming the above-mentioned results. Furthermore, our in vitro experiments (Fig. 5) also showed that TGF-β1 is indeed important for the induction of IL-22 production in already differentiated Th17 cells. The unaffected frequency of IL-17A-IL-22+ T cells in the transgenic mice with T-cell specific impaired TGF-β signaling goes in line with our in vitro results in which the absence of TGF-β1 favored the development of these cells and it might imply that these cells belong to a separate T-cell lineage than Th17 cells. This observation is further supported by the data showing that the majority of Th22 cells is not derived from Th17 cells as assessed using a fate mapping mouse model (Supplementary Fig. 6).

The lower tumor development in mice harboring TGF-β-DNR CD4+ T cells correlated with a reduced frequency of IL-17A+IL-22+ CD4+ T cells, whereas IL-17A-IL-22+ CD4+ T cells were unaffected. These results could be explained by the hypothesis that IL-22 might have different functions depending on whether it is produced alone or co-produced with IL-17A. In fact, it is known that some functions of IL-22 are further induced by the presence of IL-17A, for instance, the induction of anti-microbial peptides[13,27]. However, an alternative explanation would be that the source of IL-22 does not matter and that the observed phenotype is simply owing to an overall quantitative decreased IL-22 production. Further studies are clearly warranted to test these

hypotheses. Finally, also IL-17+IL-22- T cells were reduced in mice receiving TGF-β-DNR transgenic CD4+ T-cell compared with WT. Thus, our data clearly indicate that the TGF-β signaling in CD4+ T cells and specifically Th17 cells is critical for the emergence of IL-22 producing Th17 cells and for the promotion of CRC in an IL-22-dependent manner. However, these data do not exclude a potential additional contribution of IL-17A.

Nevertheless, it is noteworthy to consider that mice also had reduced frequency of Fopx3+ T_reg, which might also influence tumor development in this setting. To exclude this, we used IL-17A^Cre × TGFBR2^fl/fl mice. These mice also showed reduced tumorigenesis compared with their wild-type littermate controls. Taken together, our results demonstrate that TGF-β directly promotes the emergence of IL-17+IL-22- and IL-17A+IL-22+ CD4+ T cells and tumorigenesis in vivo.

On a transcriptional level, several factors have been proposed to control IL-22 production. AhR is required for development of Th17 cell development and it can promote IL-22 production[3,24,56]. Expression of AhR in T cells is induced by TGF-β1 in combination with IL-6[57] and indeed correlated with high *Il22* in our in vitro experiments. Rutz et al.[29] also observed an induction of AhR, in the presence of IL-6 and TGF-β1, however this condition did not lead to IL-22 single producing cells. This effect was explained by the induction of the transcription factor cMaf that mediates a direct inhibition of IL-22 transcription by binding in the proximal site of the IL-22 promoter[29]. It is believed that the activation of AhR compensates for the inhibitory effects of cMaf, which is also induced under the same conditions, to promote IL-22 expression, although the mechanisms are still unclear. Our results indicate that strong TCR stimulation is required for the acquisition of IL-22 in T cells. Indeed, the addition of CD28 promoted the expression of AhR while suppressing cMaf. One could thus hypothesize that TCR engagement controls the balance between AhR and cMaf, therefore inhibiting or promoting IL-22 expression in the presence of TGF-β1, which might explain the controversy of the previous studies analyzing IL-22 induction.

Overall, we elucidated the regulation of IL-22 production in T cells and its impact on CRC. In vitro experiments allowed us to demonstrate that TGF-β1, with AhR ligand and strong TCR stimulation, promotes the development of IL-17+IL-22+ T cells. Furthermore, using transgenic mice with impaired TGF-β signaling in T cells and IL-17A+ cells, we could demonstrate that TGF-β is important for the emergence of IL-17+IL-22+, but not IL-17-IL-22+ T cells during tumorigenesis in vivo.

## Methods
**Human samples**. Normal adjacent colonic tissue and tumor samples were taken from CRC patients undergoing colectomy at University Medical Center Hamburg-Eppendorf. All human studies were approved by the local ethical committee

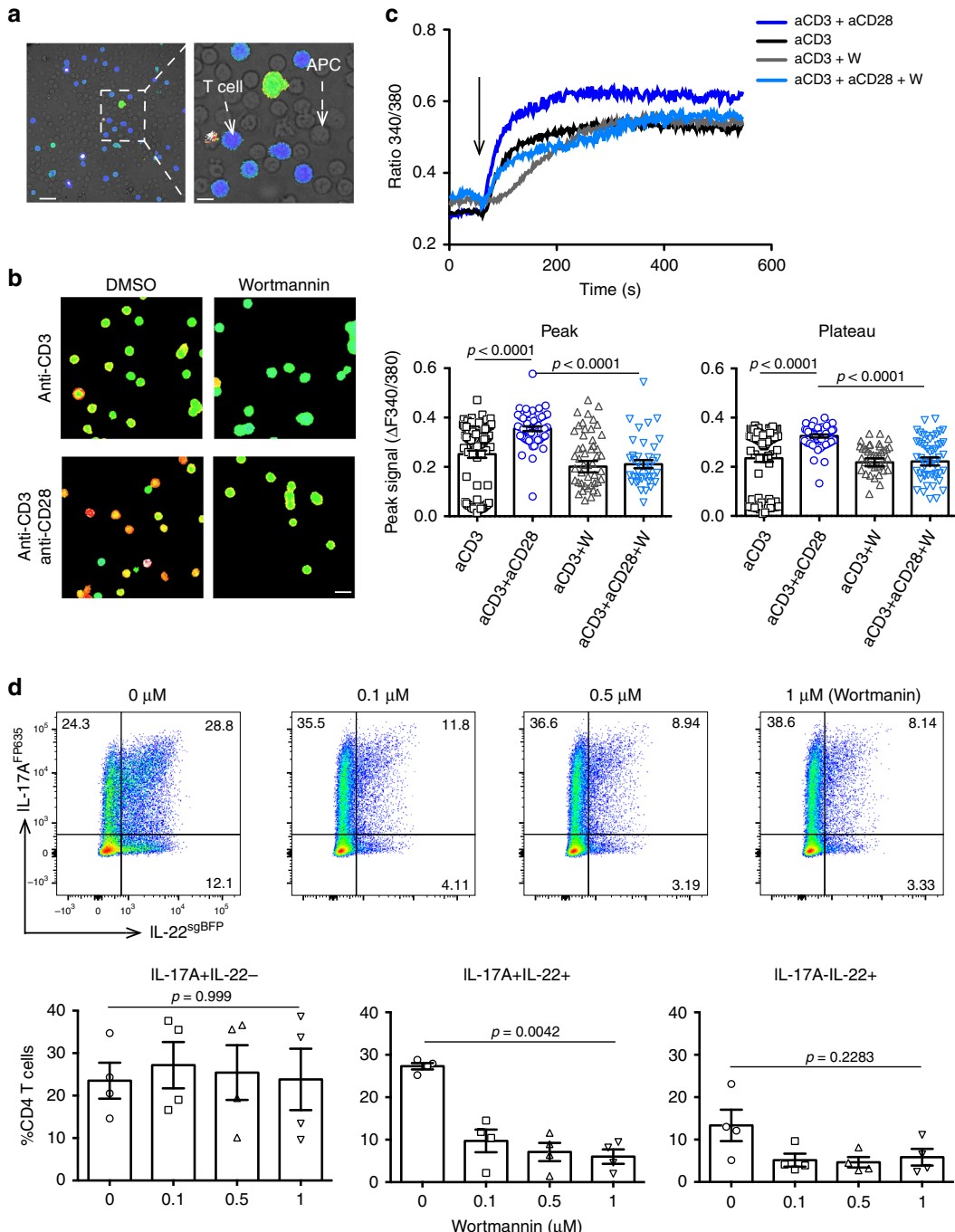

**Fig. 6 PI3 kinase mediates the differentiation of IL-17+IL-22+ T cells. Total CD4+ T cells and APCs were isolated from spleen of wild-type mice. a** Representative image of the overlay of Fura-2-labeled CD4+ T cells and unlabeled, representative from three independent experiments. Scale bar represents 15 μM (left) and 5 μM (right). **b** Representative calcium signal of Fura-2-labeled CD4+ T cells after TCR stimulation in the presence or absence of the PI3K inhibitor (Wortmanin), representative from three independent experiments. Scale bar represents 10 μM **c** Calcium tracing of stimulated CD4+ T cells over time in the presence or absence of the PI3K inhibitor (Wortmanin) (upper panel) and the corresponding statistics (lower panel). Each dot represents one cell, aCD3 $n = 90$, aCD3 + aCD28 $n = 53$, aCD3 + W $n = 51$, aCD3 + aCD28 + W $n = 40$, bars represent mean, error bars show±sem. Data are cumulative from three independent experiments. One-way ANOVA, Tukey's multiple comparisons test. **d** Naive CD4+ T cells from Foxp3^mRFP × IL-10^eGFP × IL-17^FP635 × IL-22^sgBFP reporter mice were differentiated under Th17-polarizing conditions and increasing amounts of PI3K inhibitor (Wortmanin). Frequency of indicated cell populations are shown, $n = 4$. Bars represent mean, error bars show±sem. Data are cumulative from four independent experiments. One-way ANOVA, Dunnett's multiple comparisons test. Source data are provided as a Source data file.

(Ethik-Kommission der Ärztekammer Hamburg). Informed consent was obtained from all participants.

**Mice**. All mice used in this study were in C57/BL6 background. *Rag1*−/− mice were purchased from Jackson Laboratories. *Il22*−/−[4], dnTGF-βR2 transgenic mice[58], TGFBR2^fl/fl [59], IL-17A^Cre [60], Foxp3^mRFP [61], IL-17A^eGFP [62], IL-17A^FP635 [44], and IL-10^eGFP [63]. All mice were cared for in accordance with the institutional review board 'Behörde für Soziales, Familie, Gesundheit und Verbraucherschutz' (Hamburg, Germany) and Yale University. Approval was granted (number 28/14 and 13/17). Mice were kept under specific pathogen free conditions, ambient

temperature of 20±2°C, humidity of 55±10% and a dark/light cycle of 12 hours. Age- and sex-matched littermates between 4 and 16 weeks were used.

**Generation of IL-22sgBFP reporter mice**. In order to examine IL-22 expression in live cells, we have generated a reporter mouse in which blue fluorescent protein (BFP) faithfully follows IL-22 expression. Superglo BFP (sgBFP), derived from green fluorescence protein (GFP), was chosen since it gives an excellent signal in flow cytometry. Furthermore, BFP was chosen as it complements the colors of the other cytokine reporters currently being developed or those that have already been developed in our laboratory. An internal ribosomal entry site (IRES)-BFP reporter was integrated into the 3' untranslated region (UTR) of the Il22 gene on chromosome 10 in order to faithfully report IL-22 expression without disruption of IL-22 expression itself. In brief, a targeting vector was generated in the plasmid pEasyFlox in which a 2.9 kb short arm of the Il22 gene was followed by an IRES (640 bp) and the gene encoding BFP (735 bp), then a floxed neomycin gene for selection and lastly, a long arm encoding the 3' end of Il22 (Supplementary Fig. 2). This vector was then transfected into LC-1 cells, an albino C57BL/6 ES cell line derived in our laboratory. Cells were selected by adding G418 and resulting clones were screened for correct orientation and insertion of the construct by PCR. One clone was selected for injection into C57BL/6 blastocysts and then injected into pseudopregnant CD-1 mice. Putative chimeric mice were screened by their white coat color and then identified by PCR of tail DNA. Chimeric males were bred to C57BL/6 females to obtain mice with a germ-line transmitted heterozygous BFP reporter gene. We further crossed heterozygotes to generate mice homozygous for the reporter.

**Primary cell culture**. Naive cells were cultured at $10^6$ cells/ml in the presence of irradiated APCs in a 1–3 ratio. Cells were cultured for 5 days in IMDM medium supplemented with 10% heat inactivated FBS, 1% penicillin and streptomycin, 1% L-glutamine and 50 nM 2-mercaptoethanol. Th17 polarization condition includes: mAb IL-4 (10 µg/ml), clone: 11B11, mAb INF-γ (10 µg/ml), clone: XMG1.2, mAb CD3 (3 µg/ml), clone: 2C11, mAb CD28 (0.5 µg/ml), clone: 37.51, IL-6 (10 ng/ml), TGF-β1 (1 ng/ml), and FICZ (100 mM). As indicated, IL-23 (20 ng/ml) and IL-1β (10 ng/ml) were added.

**Cell isolation for in vitro culture**. Naive CD4 T cells were enriched from cell suspension of spleen and lymph nodes using magnetic-activated cell sorting (MACS, Miltenyi). Cells were re-suspended in MACS buffer (1× phosphate-buffered saline (PBS), 1% fetal bovine serum (FBS), 0.5% ethylenediaminete-traacetic acid (EDTA)) containing biotinylated antibodies against CD25 (1:200, clone PC61, Biolegend) and CD44 (1:200, clone IM7, Biolegend) for 15 minutes at 4°C. Cells were washed with MACS buffer and pelleted by centrifugation ($350 \times g$, 5 minutes, 4°C). The pellet was re-suspended in MACS buffer containing Strep-tavidin Microbeads (40 µl/ml, 130-048-102, Miltenyi) for 30 minutes at 4°C. The cell suspension was run through a MACS LS column and washed three times with 3 ml of MACS buffer. The flow through with CD25-and CD44-cells was collected, pelleted ($350 \times g$, 5 minutes, 4°C) and further incubated with MACS buffer containing CD4 (L3T4) microbeads (100 µl/ml, 130-117-043, Milteny) and incubated 30 minutes at 4°C. The cell suspension was run through a MACS LS column and washed three times with 3 ml of MACS buffer. Labeled CD4+ cells, considered as naive T cells were collected after flushing the column with MACS buffer. To collect APCs, the flow through was further incubated with biotinylated CD3 antibody (1:200, clone 145-2C11, Biolegend) for 15 minutes at 4°C. Cells were washed and pelleted by centrifugation then re-suspended in MACS buffer containing Strepta-vidin microbeads (40 µl/ml). After 30 minutes of incubation at 4°C, the cell suspension was run through another MACS LS column and washed. The flow through contains the APCs that were irradiated with 30 Gy prior culture.

**Adoptive transfer**. Total CD4+ T cells were isolated from spleen and lymph nodes using MACS. Rag1−/− and Rag1−/− × Il22−/− were engrafted with two million CD4+ T cells, injected intraperitoneally, for 4–5 weeks.

**Cell preparation**. Intestinal specimens were cut into small pieces and incubated at 37°C for 20 min in DTT buffer (10% ml 10× Hank's balanced salt solution, 10% 10× HEPES-bicarbonate buffer, 10% ml fetal bovine serum heat inactivated and 1 nM dithioerythritol). After this step, the intraepithelial lymphocytes (IEL) were collected by centrifugation and the supernatant was discarded. The pieces of intestine were collected and incubated at 37°C for 45 min in Collagenase buffer (10% fetal bovine serum heat inactivated, 1% 100× HGPG, 1 mM CaCl₂, 1 mM MgCl₂, 100 U/ml collagenase and DNase in RPMI 1640) to obtain lamina propria lymphocytes (LPL). The digested intestinal tissue was further homogenized through a metal strainer and pooled to the IEL. Both fractions (IEL and LPL) were pelleted by centrifugation and further separated with a 67–40% Percoll gradient.

**C. rodentium infection**. Nalidixic acid-resistant, luciferase-expressing derivative of C. rodentium (ICC180) was grown overnight in Lysogeny broth (LB) containing 50 µg/ml of nalidixic acid with shaking at 37°C. Next day, the suspension of bacteria was washed twice with PBS and adjusted the concentration to 5 × $10^9$ cfu/ml. Mice were infected by oral gavage with 0.2 ml of C. rodentium solution containing 1 × $10^9$ cfu[64,65].

**Tumor induction**. Co-housed mice were injected intraperitoneally with 7.5 mg/kg body weight Azoxymthane (AOM, Sigma). After 5 days, mice were fed 2% DSS (MP biomedicals, M.W. = 36,000–50,000 Da) in the drinking water for 5 days, followed by 16 days of regular water. This cycle was repeated twice[8].

**Colonoscopy**. Colonoscopy was performed in a blinded fashion for tumor monitoring using the Coloview system (Karl Storz, Germany)[21]. Mice were anesthetized with Isofluran. Tumor sizes were graded from 1 to 5. The total tumor score per mouse was calculated as a summary of all tumor sizes[8].

**Ca²⁺ imaging**. Calcium-imaging was performed using murine CD4+ T cells and APCs isolated by magnetic cell sorting. After isolation, the CD4+ T cells were loaded with Fura-2/AM (Calbiochem) as described previously[66] and kept in the dark at room temperature (RT) until use. The cells were preincubated with Wortmannin (1 µM) or DMSO (0.25%) for 1 hour. For ratiometric Ca²⁺ imaging, thin glass coverslips (0.1 mm) were coated with Bovine serum albumin (5 mg/mL) and poly-L-lysine (0.1 mg/mL). A rubber ring was attached to the coverslip using silicon grease in order to create a small measurement-chamber. In all, 80 µL Calcium-measurement buffer (140 mM NaCl, 5 mM KCl, 1 mM MgSO₄, 1 mM CaCl₂, 1 mM NaH₂PO₄, 5.5 mM glucose, and 20 mM Hepes, pH 7.4), 10 µL CD4+ T-cell suspension (final concentration $1 \times 10^6$ cells/ml) and 10 µL APC suspension (final concentration $4 \times 10^6$/mL, not loaded with Fura-2/AM) dissolved in the same buffer were added into the small chamber. Ratiometric Ca²⁺ imaging was performed as described previously[67,68]. Sixty seconds after the start of the measurement, the indicated antibodies were added to the chamber (final concentrations 10 µg/ml). Data processing was performed using Openlab software (Improvision). The peak was defined as the delta between the maximal signal and the resting calcium level (defined as the average F340/F380 ratio in the first 60 seconds). The plateau was defined as the delta between the average F340/F380 ratio between 480 s and 540 s after the start of the measurement and the resting F340/F380 ratio.

**ELISA**. IL-22 levels in cell culture supernatant were measured using a mouse IL-22 ELISA kit (Antigenix America) according to the manufacturer's instructions.

TGFβ abundance in lysates from frozen human tumor and control tissue was determined by TGFβ1 $E_{max}$ ImmunoAssay System (Promega) according to the manufacturer's instruction. To measure active TGFβ1, lysates were activated with 1 N HCl.

**CBA (cytometric bead array)**. For supernatant cytokine quantification LEGEN-Dplex mouse T helper cytokine panel was used. Cell culture supernatants were diluted with assay buffer (1:100). In a V-bottom plate 25 µL assay buffer were added to all wells, 25 µL diluted standard were added to standard wells, 25 µL of each sample were added to sample wells, 25 µL mixed beads were added to all wells. The plate was incubated for 2 hours at RT, shaking. The plate was spinned down ($250 \times g$, 5 minutes), supernatant was discarded by inverting and flicking the plate. The plate was washed one time by adding 200 µl of wash buffer. After spining and discarding supernatant 25 µL detection antibodies were added to all wells and incubated for 1 hour, at RT, shaking. Without washing, 25 µL of Streptavidin-PE were added and incubated 30 min, at RT, shaking. The plated was spinned down and 150 µL of 1× wash buffer were added to read on a flow cytometer.

**Real-time PCR**. Total RNA was isolated and reverse-transcribed using random primers. cDNA was analyzed by real-time PCR using TaqMan assays containing a set of primers and reporter probes. Hprt1 Mm03024075_m1, Il22 Mm01226722_g1, Ahr Mm00478932_m1, Il17a Mm00439618_m1, Rorc Mm01261022_m1, Il10 Mm00439614_m1, Maf Mm02581355_s1.

**Flow cytometry**. For extracellular staining cells were re-suspended in 100 µl MACS buffer (1× PBS, 1% FBS, 0.5% EDTA) containing directly fluorochrome labeled antibodies and incubated for 20 minutes at 4°C in the dark. Cells were washed and re-suspended in 300 µl of fluorescence-activated cell sorting buffer for direct acquisition or further extracellular staining. For the staining of cytokines in the human specimens we re-suspended the cells in complete medium containing PMA (50 ng/ml), Ionomycin (1 mM), and Monensin A (2 µM) and incubated them at 37°C for 2.5 hours. The cells were washed and pelleted and surface staining was performed as described above. Fixation (4% formaldehyde, 20 minutes, RT) and permeabilization (0.1% of NP40, 4 minutes, RT) was performed before the intra-cellular staining. The cells were re-suspended in 100 µl of MACS buffer containing directly labeled fluorochrome antibodies against intracellular cytokines for 1 h at 4°C.

*Mouse antibodies*. CD4-PeCy7, RM4-5, Biolegend, B249467 (1:400); TCR-B-APC, H57-597, Biolegend, B160272 (1:400); CD45.1-APC, A20, Biolegend, B209251 (1:400); CD45.2-PECy7, 104, Biolegend, B214388 (1:400); CD4-APC-Cy7, GK1.5,

Biolegend, B213762 (1:800); CD4-BV786, RM4-5, Biolegend, B236637 (1:1000); CD3-PE, 17A2, Biolegend, B210714 (1:800); CD4-BUV737, GK1.5, BD, 4324664 (1:800); CD45.2-AF700, 104, Biolegend, B252126 (1:200); CD8a APC-Cy7, 53-6.7, Biolegend, B217172 (1:600); CD3e-BUV395, 17A2, BD, 8043925 (1:800).

*Human antibodies.* IL-17A-BV421, BL168, Biolegend, B192196 (1:200); IL-22-PE, 22URTI, eBioscience, E11018-1633 (1:200); TNF-a-BV605, MAb1, Biolegend, B191202 (1:200); INF-g-BV786, 4 S.B3, Biolegend, B216665 (1:250); CD3-BUV737, UCHT1, BD, Biosciences, 7045559 (1:200); CD4-PECy7, OKT4, Biolegend, B189535 (1:600); CD45-PECy5, HI30, Biolegend, B215344 (1:400).

**Reporting summary**. Further information on research design is available in the Nature Research Reporting Summary linked to this article.

## Data availability

The authors declare that all data are available in the article file and supplementary information files or available from the authors upon reasonable request. The source data underlying Figs. 1a, c, d, 2a, b, d, 3a, b, 4a, b, 5a–d, 6c, d, and Supplementary Figs. 1, 2, 4–7 are provided as a Source Data file.

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

## Acknowledgements

We thank the FACS Sorting Core Unit of the Universtätsklinikum Hamburg-Eppendorf for their support. We thank Cathleen Haueis and Sandra Wende for their excellent technical assistance. This work was supported by the SFB1328 (to A.H.G., A.G., S.H.), SFB841 (to A.G. and S.H.), Stiftung experimentelle Biomedizin (to S.H.), DFG HU1714/9-1 (to S.H.) and ERC StG337251 (to S.H.), and the SFB-TR241 (to S.G.).

## Author contributions

S.H, N.G., and L.G.P. conceived and designed the experiments. L.G.P., J.K., P.P., T.A., A.G., L.K., L.B., R.W., H.X., and B.P.D. performed the experiments. Data interpretation and analysis was performed by L.G.P. and S.H. R.A.F. and H.M.M. generated and provided the IL-22sgBFP reporter mouse used in this study. T.B., S.S., B.S., M.C.A.V., C.M., A.H.G., and O.S contributed with important intellectual input. D. P. and J.R.I. provided the human samples. S.G. provided mice. Writing, reviewing, and manuscript editing was done by L.G.P., S.H., and R.A.F.

## Competing interests

The authors declare no competing interests.
