## [Peer Review File · Nature Communications]

Reviewers' comments:

Reviewer #1, expertise in IL-17/IL-22 and inflammation-driven CRC (Remarks to the Author):

The manuscript by Perez et al reported a novel mechanism by which TGF- β promotes IL-22 production from Th17 cells and therefore augments the development of colitis associated colorectal cancer. The authors performed elegant studies showing that: 1), IL-17/IL-22 double positive CD4+ T cells are enriched in human colorectal cancers; 2), TGF- β signaling to CD4+ T cells, and specifically, to differentiated Th17 cells, promotes the production of IL-22; 3), this direct TGF- β signaling in Th17 cells for the production of IL-22 is dependent on strong TCR stimulation and expression of AhR; and 4), TCR engagement induces Ca²⁺ signaling that is dependent on PI3K pathway, and that the blockade of PI3K signaling reduced the expression of IL-22 in Th17 cells. These conclusions were backed up by a series of in vitro and in vivo experiments, especially the use of multiple lineage tracing mouse models for the direct visualization of IL-17, IL-22, and IL-10-expressing cells. Overall the work was well done, and the function of TGF- β in differentiated Th17 cells that promotes IL-22 production and CAC development is of great interest.

There are however several points that this reviewer would like to invite the authors' attention:

1. The authors discussed several times in the manuscript that IL-22 has bi-phasic roles in cancer development: on one hand, IL-22 protects that host against genotoxic stress; on the other hand, IL-22 promotes cancer development in mice and therapy resistance in humans. In the manuscript, the authors proposed that it may matter if IL-22 is produced alone or together with IL-17A (Discussion section, 6th paragraph). However, given the soluble nature of IL-22 and IL-17A, it is hard to imagine that IL-22 secreted from IL-17-producing Th17 cells function differently from IL-22 that originate from Th22 or innate immune cells. The authors may consider clarifying this speculation.
2. Figure 1B showed that there is a trend of increased IL-22+ cells among CD4+ T cells in human colorectal tumors, but the difference was not statistically significant. The authors may want to tune down the claim in the text that there is an increase in such cells. In fact, it is more critical, as the authors showed subsequently, that the proportion of IL-17A+ IL-22+ CD4+ T cells increased significantly in human CRC (Figures 1C, D).
3. Similarly, the percentage of regulatory T cells (Tregs) was decreased in normal colon tissue but not in tumors of CAC model (Fig S4). While this may not be the main point of this study, one can benefit from clearer description of the data in the text section.
4. The authors showed that when TGF- β signaling was blocked in T cells (Figure 3) and differentiated Th17 cells (Figure 4), there were both reduced CAC development in mice. While these data are intriguing and show for the first time the requirement of continued TGF- β signaling in Th17 cells for the promotion of CAC, one cannot rule out the possibility that the reduced CAC development was due to the reduction in IL-17 production. Indeed, as the authors showed in Figure 3A and 4A, TGF- β signaling to T cells, and specifically, differentiated Th17 cells, promotes the expression of IL-17A. IL-17A is known to promote tumor development in the AOM/DSS model.
5. In Figure 5A, the levels of IL-10 appear similar between groups of weak v.s. strong TCR stimulation. Is the difference significant?
6. The authors claimed that TGF- β induced the expression of AhR and thus IL-22 production in Th17 cells. The data showed that AhR is induced upon stimulation of naïve T cells with both IL-6 and TGF- β . Is TGF- β alone effective in AhR induction? Could the authors observe increased AhR expression when differentiated Th17 cells were stimulated with TGF- β (as shown in Figure 5C)?
7. In the text, the authors hypothesized that TGF- β may promote IL-22 production via the induction of PI3K pathway together with TCR engagement. However, Figure 6 only showed the role of strong TCR stimulation in the activation of PI3K/Ca²⁺ pathway. Is TGF- β signal still required for PI3K activation of Ca²⁺ influx? Since the T cells were stimulated with strong TCR engagement (which favors IL-22 production in these cells), why do they still require TGF- β to activate the same pathway?

Reviewed by Kepeng Wang

Reviewer #2, expertise in IL-22 (Remarks to the Author):

The role of TGF-beta in the production of IL-22 in CD4(+) T cells still remains controversial. To clarify this important issue both scientifically and clinically, this manuscript employs a series of reporter mouse systems. The reporter mouse systems include triple reporter mice for detection of IL-17A, IL-22, and Foxp3 and fate-map double reporter mice for identifying IL-22-producing cells without versus with prior IL-17A expression. The authors initially confirm the increase of TGF-beta expression and IL-17A(+) IL-22(+) T cells in the tissue of human colorectal cancer. In vitro experiments using mouse T cells reveal an ability of TGF to promote the emergence of IL-17A(+) IL-22(+) CD4(+) T cells. In vivo experiments using T cells in which TGF-beta signal is impaired due to the overexpression of dominant negative TGF-betaRII as well as T cells in which TGFbetaR2 is specifically deleted in Th17 cells (under control of IL-17A promoter) demonstrate that TGF-beta induces the differentiation of IL-17(+)IL-22(-) and IL-17(+)IL-22(+), but not IL-17(-)IL-22(+), T cells. The contribution of IL-17(+)IL-22(+) T cells for promoting carcinogenesis pathway are also shown by transferring T cells with intact versus impaired TGF-beta signaling into immune-deficient colitis-associated cancer model without or with endogenous IL-22. A series of in vitro experiments propose that, in addition to TGF-beta signaling, Ahr activation and strong TCR signaling mediated by PI3K are required for the differentiation of IL-17(+)IL-22(+) T cells. Based on these data, the authors conclude that TGF-beta signaling in Th17 cells promotes IL-22 production and colitis associated cancer. Overall, this manuscript is carefully designed with a substantial depth to minimize potential problems and provide a novel insight into Th17 biology.

Here are some specific comments:

The authors emphasize colitis-associated cancer throughout the manuscript. However, it is unclear whether colitis-associated cancer or sporadic cancer was used for the analysis of human samples.

The average of IL-17A(+)IL-22(+) in the control (WT/WT) shown in Fig3A is approximately 3%. In contrast, the average of IL-17A(+)IL-22(+) in the control (WT/TGFB^{Rwt/wt}) shown in Fig4A is approximately 0.3%. The authors need to clarify why there are 10 times-difference in the control from a figure to a figure.

It becomes increasingly apparent that CD4(+) T cells expressing both IL-17 and Foxp3 develop particularly in the intestine (Immunity 2019, p212; Nat Immunol 2019, p471). Since the authors used a triple reporter mouse system capable of detecting not only IL-17A and IL-22 but also Foxp3, it may be appreciated if they show whether IL-17(+) IL-22(+) T cells express Foxp3 or not.

Reviewer #3, expertise in T helper cell differentiation (Remarks to the Author):

The authors of this manuscript aimed to resolve some of the controversy surrounding the role of TGF-β mediated regulation of IL-22 production in CD4 T cells. Using transgenic reporter mice for IL17A and IL22 they clearly demonstrate in vitro that TGF-β and strong TCR stimulation coupled with AhR ligands, promotes the emergence of IL-17+IL-22+ T cells as well as the production of IL-22 in already differentiated Th17 cells. They relate these results to in vivo models of intestinal tumorigenesis demonstrating impaired TGF-β signaling in T cells reduces IL-22 production in Th17 cells and subsequent tumor burden. Finally, they show both IL-17A+IL-22+ producing T cells as well as TGF-β levels are increased in human CRC samples compared to normal adjacent tissue suggesting the conclusions drawn from their in vitro and mouse models are also likely true in the context of human disease. However, the manuscript lacks the evidence showing a causal effect of IL-17+IL-22+ cells and the development of colorectal cancer and the roles of IL-17+IL-22- or IL-17-IL-22+ in

tumorigenesis are unclear.

For the AOM/DSS CRC mouse model, the authors transferred congenically marked wild type or TGF- β -DNR transgenic (Tg) CD4+ T cells into Rag1-/- mice. This is concerning because of the lymphopenic niche in Rag1-/- mice will lead to homeostatic proliferation of the transferred cells. It is important to determine if the same results would be achieved if the cells were transferred into mice with fully intact adaptive immune systems.

In Figure 5 in addition to using Ahr inhibitor, the data need to be substantiated by using AhR-deficient CD4+ T cells.

Other comments:

From the methods, MACS kits were used for cell isolations but no post-enrichment purity was shown. In general, FACS data on IL-22 expression in transfer models need improvement. It needs special attention for *Citrobacter rodentium* model in which IL-22 can barely be detected.

Comments by the editor

We hope you will find the referees' comments useful as you decide how to proceed. Should further experimental data or analysis allow you to address these criticisms within six months, we would be happy to look at a substantially revised manuscript. However, please bear in mind that we would not consider the manuscript for publication in the absence of major revisions.

We are grateful for the overall positive assessment of our manuscript and for the opportunity to submit a revised version. The reviewer's comments have helped to further improve our study and we provide a point by point reply of our revisions below.

Specifically, the revisions must include (but are not limited to):

- 1. Addressing all technical concerns of our referees, including statistical analyses and variability of IL-22+ IL-17+ cell frequency across experiments; please also provide absolute numbers.**

As requested, we have included the statistical analysis and absolute cell numbers.

As for the variability of IL-17A+IL-22+ T cells, we agree with the reviewer that we observed fewer IL-17A and IL-22 producing cells in the experiment shown in Figure 4 compared to the experiment shown in Figure 3. This effect was most prominent for the IL-17A+ IL-22+ double positive cells, but the IL-17 and IL-22 single producing cell frequencies were also lower.

We would, however, like to point out that this was not due to a technical problem because the isolation (we had similar cell numbers and similar amounts of living cells), staining, and analysis were similar between both sets of experiments.

Of note, the experiments shown in Figure 3 and Figure 4 were not performed as a head to head comparison, but over the course of five years. Indeed, the experiments using the TGF- β -DNR transgenic mice were performed in 2015, while the experiments using the IL-17A^{Cre} x TGFBR2^{Flox/Flox} within the last year. During this time our mice, including the *Rag1*^{-/-} mice, were moved to another breeding facility and the diet was also changed. Of course, this also had a major impact on the intestinal microbiota. Indeed, we are currently analyzing the role of different intestinal microbiota compositions on the emergence of IL-17+IL-22+ producing cells and have found a major role of SFB on the emergence of IL-22 producing cells (independent of the known effects on IL-17A production). However, we believe that the study of these environmental effects goes beyond the scope of this manuscript.

However, since we were aware of the potentially high impact of the intestinal microbiota, we intentionally used littermate controls and co-transfer systems for the mouse *in vivo* experiments. This allowed us not only to discriminate between cell extrinsic and cell intrinsic effects, but also to control for the intestinal microbiota.

In light of this, we now discuss this in the revised version of the manuscript:

'Indeed, we did observe a high variability in the frequency of IL-17+IL-22+ CD4+ T cells over the course of the in vivo experiments. This effect might be due to different intestinal microbiota compositions of the mouse lines used. We therefore used littermate controls and co-transfer experiments in order to control for microbial effects.' (page: 13, line: 453)

2. Causative evidence that TGF- β signaling in Th17 cells promotes tumorigenesis by inducing IL-22; or more generally, functional evidence that the T-cell source of IL-22 impacts physiology or pathology. We realize that the second point requires long-term in vivo experiments with complex genetics, and that their results may not support the hypothesis that IL-22 plays a distinct and critical role in IL-17-producers. If this turns out to be the case, we can consult with our colleagues at Communication Biology on whether the degree of advance would be sufficient for them to offer publication.

We do agree that this is a key point, and as a result, we have indeed performed several additional experiments which support the note that *'TGF- β signaling in Th17 cells promotes tumorigenesis by inducing IL-22'*.

First, we found that TGF- β signaling in CD4+ T cells is important for the emergence of IL-22 producing Th17 cells (Figure 1 and 3). Thus, impaired TGF-beta signaling led to a reduction of IL-22 producing Th17 cells and correlated with a decreased tumor load in the colon (Figure 3). Of note, IL-22 single producing T cells were not affected by the loss of TGF- β signaling.

Second, we found that TGF- β signaling, specifically in Th17 cells promotes IL-22 production *in vitro* (Figure 5) and *in vivo* (Figure 4). Accordingly, mice with a Th17 cell specific blockade of TGF-beta signaling showed reduced tumorigenesis in the colon (Figure 4).

Third, in order to finally prove that TGF- β signaling promotes tumorigenesis by inducing IL-22 producing Th17 cells, we repeated the experiment shown in Figure 3 in an IL-22 free environment. To this end, we transferred *Il22*^{-/-} and TGF- β -DNR transgenic x *Il22*^{-/-} CD4+ T cell into *Rag1*^{-/-} x *Il22*^{-/-} mice. As control, we transferred wild type and TGF- β -DNR transgenic

CD4⁺ T cells. Indeed, we could confirm that mice receiving wild type T cells showed a higher tumor load compared to mice receiving TGF- β -DNR transgenic CD4⁺ T cells (new Figure 3, below). This effect cannot be due to a difference in IL-22 single producing cells, as we found similar frequencies of these cells in both groups (Figure 3A). Of note, in an IL-22 free environment mice receiving *Il22*^{-/-} or TGF- β -DNR transgenic x *Il22*^{-/-} CD4⁺ T cell showed an equal tumor load (new Figure 3, below) indicating that the observed effect is indeed IL-22 dependent.

Taken together, these experiments show that TGF- β signaling in CD4⁺ T cells and specifically, Th17 cells is in fact critical for the emergence of IL-22 producing Th17 cells and for the promotion of colorectal cancer in an IL-22 dependent manner.

Fig. 3: TGF- β signaling promotes the emergence of IL-17+IL-22+ T cells in a direct manner *in vivo*. **A)** Congenic CD4⁺ T cells from Foxp3^{mRFP} x IL-17A^{eGFP} x IL-22^{sgBFP} or Foxp3^{mRFP} x IL-17A^{eGFP} x IL-22^{sgBFP} x dnTGF- β 2 (Tg) mice were co-transferred into *Rag1*^{-/-} prior to tumor induction using AOM/DSS. Production of IL-17A and IL-22 by T cells was analyzed in tumors and normal adjacent tissue (control) using flow cytometry. Results are cumulative from two independent experiments. Control: (WT:WT) n= 11; (WT:Tg) n= 12. Tumor: (WT:WT) n=9; (WT:Tg) =9. Lines indicate mean +/- sem. Wilcoxon multiple comparisons test was performed (P<0.05) to assess the significance. **B)** Colitis associated colon cancer was induced in *Rag1*^{-/-}/*Il22*^{-/-} mice upon reconstitution with wild type (WT), dnTGF- β 2 (Tg), *Il22*^{-/-} or dnTGF- β 2 x *Il22*^{-/-} (Tg x *Il22*^{-/-}) CD4⁺ T cells. Tumor score and tumor number is shown. **C)** Representative endoscopic view and **D)** histological sections are shown; arrows indicate tumors; scale bar represents 500 μm. Results are cumulative from two independent experiments. Each dot represents one mouse (WT n= 18; Tg n= 14; *Il22*^{-/-} n= 19; Tg x *Il22*^{-/-} n= 7). Lines indicate mean +/- sem; Tukey's multiple comparisons test was performed (P<0.05) to assess the significance. Source data are provided as a Source data file.

- 3. Acknowledgement and discussion of the relevant most current literature, including doi: 10.1016/j.immuni.2019.05.004. This particular article, in our view, considerably dents the novelty of phenotypic characterization of the intestinal IL17 and IL-22 double and single producers. We therefore encourage you to focus the revisions on the aspects that advance the manuscript beyond the reported findings.**

We now acknowledge and discuss the relevant most current literature throughout the manuscript including doi: 10.1016/j.immuni.2019.05.004 and we focus on the aspects that advance the manuscript beyond the reported findings. We apologize for the omission of the above mentioned very important manuscript, which was published (16th July 2019) briefly after we submitted this manuscript (8th July 2019) and thank you for pointing this out. This manuscript provides a very nice phenotypical and functional characterization of homeostatic and infection induced Th17 cells. As for IL-22, *Omenetti et al.* found that Th17 cells co-produce IL-22 upon SFB-colonization (homeostatic condition) and upon *Citrobacter rodentium* infection. However, this study did not focus on the mechanism(s) regulating IL-22 production by Th17 cells, nor did it focus on the impact of IL-22 on colorectal cancer. Thus, while this manuscript provides an important description of different Th17 cell subsets in different settings, it does not compromise the novelty of our finding that TGF- β promotes IL-22 production by Th17 cells and therefore augments the development of colitis associated colorectal cancer. We have, however, included a discussion of this study in the revised manuscript:

'Interestingly, Th17 cells have been described to produce IL-22 in homeostatic conditions and upon Citrobacter rodentium infection [69]. However, the specific function of these cells in CRC and the mechanisms regulating them remained elusive. Furthermore, it was unclear whether IL-17A and IL-22 are co-produced by the same cells or by different cells and unclear which factors would regulate IL-22 expression in the respective cells. These points are critical when it comes to designing new therapies targeting IL-17A and IL-22.' (page:11, line: 377)

Reviewer: 1

The manuscript by Perez et al reported a novel mechanism by which TGF- β promotes IL-22 production from Th17 cells and therefore augments the development of colitis associated colorectal cancer. The authors performed elegant studies showing that: 1), IL-17/IL-22 double positive CD4⁺ T cells are enriched in human colorectal cancers; 2), TGF- β signaling to CD4⁺ T cells, and specifically, to differentiated Th17 cells, promotes the production of IL-22; 3), this direct TGF- β signaling in Th17 cells for the production of IL-22 is dependent on strong TCR stimulation and expression of AhR; and 4), TCR engagement induces Ca²⁺ signaling that is dependent on PI3K pathway, and that the blockade of PI3K signaling reduced the expression of IL-22 in Th17 cells. These conclusions were backed up by a series of in vitro and in vivo experiments, especially the use of multiple lineage tracing mouse models for the direct visualization of IL-17, IL-22, and IL-10-expressing cells. Overall the work was well done, and the function of TGF- β in differentiated Th17 cells that promotes IL-22 production and CAC development is of great interest.

We thank this reviewer for the very positive assessment of our work and for recognizing the novelty of our study.

Major points:

1. The authors discussed several times in the manuscript that IL-22 has bi-phasic roles in cancer development: on one hand, IL-22 protects that host against genotoxic stress; on the other hand, IL-22 promotes cancer development in mice and therapy resistance in humans. In the manuscript, the authors proposed that it may matter if IL-22 is produced alone or together with IL-17A (Discussion section, 6th paragraph). However, given the soluble nature of IL-22 and IL-17A, it is hard to imagine that IL-22 secreted from IL-17-producing Th17 cells function differently from IL-22 that originate from Th22 or innate immune cells. The authors may consider clarifying this speculation.

We agree with this reviewer and -as suggested- clarified this speculation:

'The lower tumor development in mice harboring TGF- β -DNR CD4⁺ T cells correlated with a reduced frequency of IL-17A+IL-22⁺ CD4⁺ T cells, while IL-17A-IL-22⁺ CD4⁺ T cells were unaffected. These results could be explained by the hypothesis that IL-22 might have different functions depending on whether it is produced alone or co-produced with IL-17A. In fact, it is known that some functions of IL-22 are further induced by the presence of IL-17A, for instance, the induction of anti-microbial peptides [16, 35]. However, an alternative explanation would be

that the source of IL-22 does not matter and that the observed phenotype is simply due to an overall quantitative decreased IL-22 production. Further studies are clearly warranted to test these hypotheses.' (page: 13, line: 478)

- 2. Figure 1B showed that there is a trend of increased IL-22+ cells among CD4+ T cells in human colorectal tumors, but the difference was not statistically significant. The authors may want to tune down the claim in the text that there is an increase in such cells. In fact, it is more critical, as the authors showed subsequently, that the proportion of IL-17A+ IL-22+ CD4+ T cells increased significantly in human CRC (Figures 1C, D).**

We agree and we modified the text accordingly:

'The analysis of fresh colorectal cancer specimens indicated an increased infiltration of CD4+ T cells producing IL-22 by trend compared to adjacent normal colon tissue (Frequency of IL-22 producing cells within CD4+ T cells: Control: 5.09+/-5.5 vs Tumor: 7.68+/-6.21, p= 0.0879). Further characterization of these cells showed that the increased IL-22 frequency in the tumor was due to an enrichment of IL-17A+IL-22+ double producing CD4+ T cells, while IL-17A-L-22+ single producing T cells were not changed (Figure 1B, C, D).' (page: 5, line: 174)

- 3. Similarly, the percentage of regulatory T cells (Tregs) was decreased in normal colon tissue but not in tumors of CAC model (Fig S4). While this may not be the main point of this study, one can benefit from clearer description of the data in the text section.**

We agree and we modified the text accordingly:

'As expected [24], the frequency of Foxp3+ CD4+ T cells (Figure S4) was also reduced in TGF- β -DNR transgenic CD4+ T cells compared to wild type control in normal colon. However, this was not the case in the tumor tissue.' (page: 7, line: 236).

- 4. The authors showed that when TGF- β signaling was blocked in T cells (Figure 3) and differentiated Th17 cells (Figure 4), there were both reduced CAC development in mice. While these data are intriguing and show for the first time the requirement of continued TGF- β signaling in Th17 cells for the promotion of CAC, one cannot rule out the possibility that the reduced CAC development was due to the reduction in IL-17 production. Indeed, as the authors showed in Figure 3A and 4A, TGF- β signaling to T cells, and specifically, differentiated Th17 cells, promotes the expression of IL-17A. IL-17A is known to promote tumor development in the AOM/DSS model.**

We agree that this is a key point and we have thus performed further experiments to clarify this aspect. Indeed, we now provide several lines of evidence supporting the note that TGF- β signaling in Th17 cells promotes tumorigenesis by inducing IL-22.

First, we found that TGF- β signaling in CD4+ T cells is important for the emergence of IL-22 producing Th17 cells (Figure 1 and 3). Thus, impaired TGF-beta signaling led to a reduction of IL-22 producing Th17 cells and correlated with a decreased tumor load in the colon (Figure 3). Of note, IL-22 single producing T cells were not affected by the loss of TGF- β signaling.

Second, we found that TGF- β signaling specifically in Th17 cells promotes IL-22 production *in vitro* (Figure 5) and *in vivo* (Figure 4). Accordingly, mice with Th17 cell specific blockade of TGF-beta signaling showed reduced tumorigenesis in the colon (Figure 4).

Third, in order to finally prove that TGF- β signaling promotes tumorigenesis by inducing IL-22 producing Th17 cells, we repeated the experiment shown in Figure 3 in an IL-22 free environment. To this end we transferred *Il22*^{-/-} and TGF- β -DNR transgenic x *Il22*^{-/-} CD4+ T cell into *Rag1*^{-/-} x *Il22*^{-/-} mice. As control we transferred wild type and TGF- β -DNR transgenic CD4+ T cells. Indeed, we could confirm that mice receiving wild type T cells showed a higher tumor load compared to mice receiving TGF- β -DNR transgenic CD4+ T cells (new Figure 3, below). This effect cannot be due to a difference in IL-22 single producing cells, as we found similar frequencies of these cells in both groups (Figure 3A). Of note, in an IL-22 free environment mice receiving *Il22*^{-/-} or TGF- β -DNR transgenic x *Il22*^{-/-} CD4+ T cells showed an equal tumor load (new Figure 3, below), suggesting that the observed effect is IL-22 dependent.

In conclusion, these experiments show that TGF- β signaling in CD4+ T cells and specifically, in Th17 cells is critical for the emergence of IL-22 producing Th17 cells and for the promotion of colorectal cancer in an IL-22 dependent manner. Of note, these data do not exclude a potential additional contribution of IL-17A, and we have discussed this point in the discussion of the revised manuscript.

*'Next, we analyzed whether T-cell specific impairment in TGF- β -signaling influences tumorigenesis. In order to exclude the effect of non-T-cell derived IL-22, which has been recently reported to have a protective function in CRC [8] and also to exclude effects of TGF- β on CD8+ T cells, we repopulated *Il22*-deficient lymphopenic hosts (*Rag1*^{-/-}/*Il22*^{-/-}) with either wild type (WT), TGF- β -DNR transgenic (Tg), *Il22*^{-/-} or TGF- β -DNR transgenic x *Il22*^{-/-} (Tg x *Il22*^{-/-}) CD4+ T cells and induced colitis associated colon cancer upon engraftment using the AOM/DSS model. Mice receiving wild type T cells showed a higher tumor load compared to mice receiving TGF- β -DNR transgenic CD4+ T cell (Figure 3). Moreover, in an IL-22 free*

environment mice receiving *Il22*^{-/-} or *Tg/Il22*^{-/-} CD4⁺ T cell showed an equal tumor load (Figure 3) indicating that the observed effect is IL-22 dependent.’ (page: 7, line: 244)

‘Thus, our data clearly indicate that the TGF- β signaling in CD4⁺ T cells and specifically Th17 cells is critical for the emergence of IL-22 producing Th17 cells and for the promotion of colorectal cancer in an IL-22 dependent manner. However, these data do not exclude a potential additional contribution of IL-17A.’ (page: 14, line: 488)

Fig. 3: TGF- β signaling promotes the emergence of IL-17+IL-22+ T cells in a direct manner *in vivo*. **A)** Congenic CD4+ T cells from Foxp3^{mRFP} x IL-17A^{eGFP} x IL-22^{sgBFP} or Foxp3^{mRFP} x IL-17A^{eGFP} x IL-22^{sgBFP} x dnTGF- β R2 (Tg) mice were co-transferred into *Rag1*^{-/-} prior to tumor induction using AOM/DSS. Production of IL-17A and IL-22 by T cells was analyzed in tumors and normal adjacent tissue (control) using flow cytometry. **Results are cumulative from two independent experiments. Control: (WT:WT) n= 11; (WT:Tg) n= 12. Tumor: (WT:WT) n=9; (WT:Tg) =9. Lines indicate mean +/- sem. Wilcoxon multiple comparisons test was performed (P<0.05) to assess the significance.** **B)** Colitis associated colon cancer was induced in *Rag1*^{-/-}/*Il22*^{-/-} mice upon reconstitution with wild type (WT), dnTGF- β R2 (Tg), *Il22*^{-/-} or dnTGF- β R2 x *Il22*^{-/-} (Tg x *Il22*^{-/-}) CD4+ T cells. Tumor score and tumor number is shown. **C)** Representative endoscopic view and **(D)** histological sections are shown; arrows indicate tumors; scale bar represents 500 μ m. Results are cumulative from two independent experiments. Each dot represents one mouse (WT n= 18; Tg n= 14; *Il22*^{-/-} n= 19; Tg x *Il22*^{-/-} n= 7). Lines indicate mean +/- sem; Tukey's multiple comparisons test was performed (P<0.05) to assess the significance. **Source data are provided as a Source data file.**

5. In Figure 5A, the levels of IL-10 appear similar between groups of weak v.s. strong TCR stimulation. Is the difference significant?

We agree that the difference in IL-10 levels is small. However, it is still significant ($p < 0.0001$). We have now included a statistical analysis in Figure 5A.

6. The authors claimed that TGF- β induced the expression of AhR and thus IL-22 production in Th17 cells. The data showed that AhR is induced upon stimulation of naïve T cells with both IL-6 and TGF- β . Is TGF- β alone effective in AhR induction? Could the authors observe increased AhR expression when differentiated Th17 cells were stimulated with TGF- β (as shown in Figure 5C)?

We did not study the effect of TGF- β alone on the induction of AhR by naïve T cells, since this has been done previously. Kimura *et al.* demonstrated that TGF- β alone is not sufficient for the induction of AhR expression in naive T cells (Kimura *et al.* PNAS 2008). However, we now do mention this study now in our manuscript:

'Expression of AhR in T cells is induced by TGF- β 1 in combination with IL-6 [75] and indeed correlated with high Il22 in our in vitro experiments.' (page: 14, line: 501).

Regarding the second question concerning the increased AhR expression, we tested now whether TGF- β increased AhR expression in already differentiated Th17 cells. To that end, we differentiated naïve CD4+ T cells from Foxp3^{mRFP} x IL-17A^{eGFP} x IL-22^{sgBFP} reporter mice under Th17 polarizing condition. After 4 days of culture, we sorted Th17 and re-cultured them in the presence of mAb CD3 (3 μ g/ml), mAb CD28 (0.5 μ g/ml), APCs and either TGF- β alone, or blocking TGF- β antibody or TGF- β + IL-6 as positive control. After three days, we measured AhR mRNA expression. Our results indicate that the expression of AhR in Th17 cells is maintained by the addition of TGF- β to the culture but is not further increased compared to

AhR expression in differentiated Th17 cells before the re-culture. Interestingly, AhR expression decreased significantly in the presence of the TGF- β blocking antibody. These data suggest that the TGF- β , which is present in the culture is sufficient and necessary to maintain AhR expression in differentiated Th17 cells. Finally, the addition of TGF- β and IL-6 to already differentiated Th17 cells did not further increase AhR expression (Reply Figure 1). We would be happy to include these data in the manuscript upon request.

Reply Figure 1: TGF- β maintains AhR expression in *in vitro* differentiated Th17 cells.

Naïve T cells were isolated from spleen and lymph nodes of Foxp3^{mRFP} x IL-17A^{eGFP} x IL-22^{sgBFP} reporter mice and cultured for four days under Th17 polarizing condition (mAb IL-4 (10 μ g/ml), mAb INF- γ (10 μ g/ml), mAb CD3 (3 μ g/ml), mAb CD28 (0.5 μ g/ml), IL-6 (10 ng/ml), TGF- β 1 (1 ng/ml), FICZ (100mM) and APCs). After 4 days, Th17 cells were FACS-sorted and re-cultured in the presence of mAb CD3 (3 μ g/ml), mAb CD28 (0.5 μ g/ml), APCs and TGF- β 1 (1 ng/ml), IL-6 (10 ng/ml) or neutralizing TGF- β mAb (10 μ g/ml). AhR mRNA expression of Th17 cells was measured before and after the re-culture. Bars represent mean, error bars show +/- sem. One-way ANOVA, Dunn's multiple comparisons test was performed ($P < 0.05$) to assess significance. Data are cumulative from three independent experiments.

7. In the text, the authors hypothesized that TGF- β may promote IL-22 production via the induction of PI3K pathway together with TCR engagement. However, Figure 6 only showed the role of strong TCR stimulation in the activation of PI3K/Ca²⁺ pathway. Is TGF- β signal still required for PI3K activation of Ca²⁺ influx?

TGF- β signal is not required for PI3K activation of Ca²⁺ influx. Indeed, as shown in Figure 6 A, B, C, activation of PI3K and Ca²⁺ influx was seen in the absence of TGF- β . Of note in the performed experiments, we used a calibration buffer to resuspend the cells, which is serum free and therefore also TGF- β free. Thus, TGF- β signaling is not essential for PI3K activation and Ca²⁺ influx.

- **Since the T cells were stimulated with strong TCR engagement (which favors IL-22 production in these cells), why do they still require TGF- β to activate the same pathway?**

Our data indicate that TGF- β signaling is important for Ahr induction, which then is essential to induce IL-22 production (Figure 5).

We have modified the text to clarify this point:

'Strong TCR signaling correlated with higher and longer accumulation of intracellular calcium during the early activation state, mediated by PI3K. Interestingly, activation of PI3K and Ca²⁺ influx was seen in the absence of TGF- β . In the performed experiments, we used a calibration buffer to resuspend the cells, which is serum free and therefore also TGF- β free. Thus, TGF- β signaling is not essential for PI3K activation. However, TGF- β signaling is important for Ahr induction, which then is then essential to induce IL-22 production (Figure 5). Thus, TGF- β signaling in conjunction with strong TCR stimulation was important to promote the transcription of Il22.' (page: 12, line: 439)

Reviewer: 2

The role of TGF-beta in the production of IL-22 in CD4(+) T cells still remains controversial. To clarify this important issue both scientifically and clinically, this manuscript employs a series of reporter mouse systems. The reporter mouse systems include triple reporter mice for detection of IL-17A, IL-22, and Foxp3 and fate-map double reporter mice for identifying IL-22-producing cells without versus with prior IL-17A expression. The authors initially confirm the increase of TGF-beta expression and IL-17A(+) IL-22(+) T cells in the tissue of human colorectal cancer. In vitro experiments using mouse T cells reveal an ability of TGF to promote the emergence of IL-17A(+) IL-22(+) CD4(+) T cells. In vivo experiments using T cells in which TGF-beta signal is impaired due to the overexpression of dominant negative TGF-betaRII as well as T cells in which TGFbetaR2 is specifically deleted in Th17 cells (under control of IL-17A promoter) demonstrate that TGF-beta induces the differentiation of IL-17(+)IL-22(-) and IL-17(+)IL-22(+), but not IL-17(-)IL-22(+), T cells. The contribution of IL-17(+)IL-22(+) T cells for promoting carcinogenesis pathway are also shown by transferring T cells with intact versus impaired TGF-beta signaling into immune-deficient colitis-associated cancer model without or with endogenous IL-22. A series of in vitro experiments propose that, in addition to TGF-beta signaling, Ahr activation and strong TCR signaling mediated by PI3K are required for the differentiation of IL-17(+)IL-22(+) T

cells. Based on these data, the authors conclude that TGF-beta signaling in Th17 cells promotes IL-22 production and colitis associated cancer. Overall, this manuscript is carefully designed with a substantial depth to minimize potential problems and provide a novel insight into Th17 biology.

We thank this reviewer for carefully reading our manuscript and for his very positive assessment of our work.

Specific comments:

- 1. The authors emphasize colitis-associated cancer throughout the manuscript. However, it is unclear whether colitis-associated cancer or sporadic cancer was used for the analysis of human samples.**

We clarified this point as requested:

'In order to corroborate this finding, we measured the concentration of total and active TGF- β 1 in tissue lysates from colon tumor and normal adjacent tissue from a cohort of patients with sporadic CRC (Table S1).' (page:5, line: 168)

- 2. The average of IL-17A(+)IL-22(+) in the control (WT/WT) shown in Fig3A is approximately 3%. In contrast, the average of IL-17A(+)IL-22(+) in the control (WT/TGFBRwt/wt) shown in Fig4A is approximately 0.3%. The authors need to clarify why there are 10 times-difference in the control from a figure to a figure.**

Regarding the variability of IL-17A+IL-22+ T cells, the reviewer is right that we observed fewer IL-17A and IL-22 producing cells in the experiment shown in Figure 4 compared to the experiment shown in Figure 3. This effect was most prominent for the IL-17A+ IL-22+ double positive cells, but the IL-17 and IL-22 single producing cell frequencies were also lower.

We would, however, like to point out that this was not due to a technical problem because the isolation (we had similar cell numbers and similar amounts of living cells), staining, and analysis were similar between both sets of experiments.

In line with this point made by the reviewer, we also repeated the experiment shown in Figure 4 (which is based on the co-transfer of CD4+ T cells from IL-17A^{Cre} x TGFBR2^{Flox/Flox} or IL-17A^{Cre} x TGFBR2^{Wt/Wt} littermate control mice into *Rag1*^{-/-} mice) using IL-17A^{Cre} x TGFBR2^{Flox/Flox} and IL-17A^{Cre} x TGFBR2^{Wt/Wt} littermate control mice directly and thus avoiding the transfer (Reply Figure 2). This time, the frequencies of IL-17 producing and IL-22 producing cells are comparable to those shown in Figure 4 (which is also based on the use of these mice) and again lower than the ones we observed in the experiments shown in Figure

3, in which we used the TGF- β -DNR transgenic mice and littermate controls. Thus, these lower frequencies are not due to the transfer into a lymphopenic host.

Of note, the experiments shown in Figure 3, Figure 4, and Reply Figure 2 were not performed as a head to head comparison, but over the course of five years. Indeed, the experiments using the TGF- β -DNR transgenic mice were performed in 2015, while the experiments using the IL-17A^{Cre} x TGFBR2^{Flox/Flox} within the last year. During this time our mice, including the *Rag1*^{-/-} mice, were moved to another breeding facility and the diet was also changed. Of course, this also had a major impact on the intestinal microbiota. Indeed, we are currently analyzing the role of different intestinal microbiota compositions on the emergence of IL-17+IL-22+ producing cells, and have found a major role of SFB on the emergence of IL-22 producing cells (independent of the known effects on IL-17A production). However, we believe that the study of these environmental effects goes beyond the scope of this manuscript.

However, since we were aware of the potentially high impact of the intestinal microbiota, we intentionally used littermate controls and co-transfer systems for the mouse *in vivo* experiments. This allowed us not only to discriminate between cell extrinsic and cell intrinsic effects, but also to control for the intestinal microbiota between WT and transgenic cells.

In light of this, we now discuss this in the revised version of the manuscript:

'Indeed, we did observe a high variability in the frequency of IL-17+IL-22+ CD4+ T cells over the course of the in vivo experiments. This effect might be due to different intestinal microbiota compositions of the mouse lines used. We therefore used littermate controls and co-transfer experiments in order to control for microbial effects.' (page: 13, line: 453)

Reply Figure 2: TGF- β signaling in Th17 cells promotes IL-22 production.

Colitis associated colon cancer was induced in IL-17A^{Cre} x TGFBR2^{fl/fl} or IL-17A^{Cre} x TGFBR2^{wt/wt} mice. Production of IL-17A and IL-22 by T cells from colon was analyzed by flow cytometry. Each dot represents one mouse (TGFBR2^{wt/wt} n=9; TGFBR2^{Fllox/Fllox} n=7). Lines indicate mean \pm sem; Mann-Whitney test was performed to assess significance.

3. It becomes increasingly apparent that CD4(+) T cells expressing both IL-17 and Foxp3 develop particularly in the intestine (Immunity 2019, p212; Nat Immunol 2019, p471). Since the authors used a triple reporter mouse system capable of detecting not only IL-17A and IL-22 but also Foxp3, it may be appreciated if they show whether IL-17(+) IL-22(+) T cells express Foxp3 or not.

We agree and we have performed the requested analysis. These data are depicted in Reply Figure 3. We could observe that the presence of IL-17A+Foxp3+ T cells in the tumors was not affected by the impaired TGF- β signaling. Moreover, IL-17A+IL-22+Foxp3+ T cells were almost not detectable in the tumors. We would be happy to include these data in the manuscript upon request.

Reply Figure 3: The role of TGF- β signaling for the emergence of Foxp3⁺ co-producing IL-17A and IL-22 T cells in colitis associated colorectal cancer.

CD4⁺ T cells from Foxp3^{mRFP} x IL-17A^{eGFP} x IL-22^{sgBFP} or Foxp3^{mRFP} x IL-17A^{eGFP} x IL-22^{sgBFP} x dnTGF- β R2 (Tg) mice were co-transferred into *Rag1*^{-/-} prior to tumor induction. Frequency of Foxp3⁺IL-17A⁺ and Foxp3⁺IL-17A⁺IL-22⁺ CD4⁺ T cells within CD4⁺ T cells was analyzed by flow cytometry in tumors (n=9). Results are cumulative from two independent experiments. Lines indicate mean \pm sem; 2-way ANOVA, Sidak's multiple comparisons test was performed ($P < 0.05$) to assess significance.

Reviewer: 3

The authors of this manuscript aimed to resolve some of the controversy surrounding the role of TGF- β mediated regulation of IL-22 production in CD4 T cells. Using transgenic reporter mice for IL17A and IL22 they clearly demonstrate in vitro that TGF- β and strong TCR stimulation coupled with AhR ligands, promotes the emergence of IL-17⁺IL-22⁺ T cells as well as the production of IL-22 in already differentiated Th17 cells. They relate these results to in vivo models of intestinal tumorigenesis demonstrating impaired TGF- β signaling in T cells reduces IL-22 production in Th17 cells and subsequent tumor burden. Finally, they show both IL-17A⁺IL-22⁺ producing T cells as well as TGF- β levels are increased in human CRC samples compared to normal adjacent tissue suggesting the conclusions drawn from their in vitro and mouse models are also likely true in the context of human disease.

We thank this reviewer for carefully reading our manuscript.

1. However, the manuscript lacks the evidence showing a causal effect of IL-17⁺IL-22⁺ cells and the development of colorectal cancer and the roles of IL-17⁺IL-22⁻ or IL-17⁻IL-22⁺ in tumorigenesis are unclear.

We agree that this is a key point and we have thus performed further experiments to clarify this question. Indeed, we now provide several lines of evidence supporting the note that TGF- β signaling in Th17 cells promotes tumorigenesis by inducing IL-22.

First, we found that TGF- β signaling in CD4+ T cells is important for the emergence of IL-22 producing Th17 cells (Figure 1 and 3). Thus, impaired TGF-beta signaling led to a reduction of IL-22 producing Th17 cells and correlated with a decreased tumor load in the colon (Figure 3). Of note, IL-22 single producing T cells were not affected by the loss of TGF- β signaling.

Second, we found that TGF- β signaling specifically in Th17 cells promotes IL-22 production *in vitro* (Figure 5) and *in vivo* (Figure 4). Accordingly, mice with Th17 cell specific blockade of TGF-beta signaling showed reduced tumorigenesis in the colon (Figure 4).

Third, in order to finally prove that TGF- β signaling promotes tumorigenesis by inducing IL-22 producing Th17 cells, we repeated the experiment shown in Figure 3 in an IL-22 free environment. To this end we transferred *Il22*^{-/-} and TGF- β -DNR transgenic x *Il22*^{-/-} CD4+ T cell into *Rag1*^{-/-} x *Il22*^{-/-} mice. As control we transferred wild type and TGF- β -DNR transgenic CD4+ T cells. Indeed, we could confirm that mice receiving wild type T cells showed a higher tumor load compared to mice receiving TGF- β -DNR transgenic CD4+ T cells (new Figure 3 B, below). This effect cannot be due to a difference in IL-22 single producing cells, as we found similar frequencies of these cells in both groups (Figure 3A). Of note, in an IL-22 free environment mice receiving *Il22*^{-/-} or TGF- β -DNR transgenic x *Il22*^{-/-} CD4+ T cells showed an equal tumor load (new Figure 3 B, below), suggesting that the observed effect is IL-22 dependent.

In conclusion, these experiments show that TGF- β signaling in CD4+ T cells and specifically, in Th17 cells is critical for the emergence of IL-22 producing Th17 cells and for the promotion of colorectal cancer in an IL-22 dependent manner. Of note, these data do not exclude a potential additional contribution of IL-17A, and we have discussed this point in the discussion of the revised manuscript.

*'Next, we analyzed whether T-cell specific impairment in TGF- β -signaling influences tumorigenesis. In order to exclude the effect of non-T-cell derived IL-22, which has been recently reported to have a protective function in CRC [8] and also to exclude effects of TGF- β on CD8+ T cells, we repopulated *Il22*-deficient lymphopenic hosts (*Rag1*^{-/-}/*Il22*^{-/-}) with either wild type (WT), TGF- β -DNR transgenic (Tg), *Il22*^{-/-} or TGF- β -DNR transgenic x *Il22*^{-/-} (Tg x *Il22*^{-/-}) CD4+ T cells and induced colitis associated colon cancer upon engraftment using the AOM/DSS model. Mice receiving wild type T cells showed a higher tumor load compared to mice receiving TGF- β -DNR transgenic CD4+ T cells (Figure 3). Moreover, in an IL-22 free*

environment mice receiving Il22^{-/-} or Tg/Il22^{-/-} CD4⁺ T cells showed an equal tumor load (new Figure 3) indicating that the observed effect is IL-22 dependent.' (page: 7, line: 244)

'The lower tumor development in mice harboring TGF- β -DNR CD4⁺ T cells correlated with a reduced frequency of IL-17A+IL-22⁺ CD4⁺T cells, while IL-17A-IL-22⁺ CD4⁺ T cells were unaffected. These results could be explained by the hypothesis that IL-22 might have different functions depending on whether it is produced alone or co-produced with IL-17A. In fact, it is known that some functions of IL-22 are further induced by the presence of IL-17A, for instance, the induction of anti-microbial peptides [16, 35] However, an alternative explanation would be that the source of IL-22 does not matter and that the observed phenotype is simply due to an overall quantitative decreased IL-22 production. Further studies are clearly warranted to test these hypotheses. Finally, also IL-17+IL-22⁻ T cells were reduced in mice receiving TGF- β -DNR transgenic CD4⁺ T cell compared to WT. Thus, our data clearly indicate that the TGF- β signaling in CD4⁺ T cells and specifically Th17 cells is critical for the emergence of IL-22 producing Th17 cells and for the promotion of colorectal cancer in an IL-22 dependent manner. However, these data do not exclude a potential additional contribution of IL-17A.' (page: 13, line: 478)

Fig. 3: TGF- β signaling promotes the emergence of IL-17+IL-22+ T cells in a direct manner *in vivo*. **A)** Congenic CD4⁺ T cells from *Foxp3*^{mRFP} x *IL-17A*^{eGFP} x *IL-22*^{sgBFP} or *Foxp3*^{mRFP} x *IL-17A*^{eGFP} x *IL-22*^{sgBFP} x dnTGF- β 2 (Tg) mice were co-transferred into *Rag1*^{-/-} prior to tumor induction using AOM/DSS. Production of IL-17A and IL-22 by T cells was analyzed in tumors and normal adjacent tissue (control) using flow cytometry. Results are cumulative from two independent experiments. Control: (WT:WT) n= 11; (WT:Tg) n= 12. Tumor: (WT:WT) n=9; (WT:Tg) =9. Lines indicate mean +/- sem. Wilcoxon multiple comparisons test was performed ($P < 0.05$) to assess the significance. **B)** Colitis associated colon cancer was induced in *Rag1*^{-/-} mice upon reconstitution with wild type (WT), dnTGF- β 2 (Tg), *Il22*^{-/-} or dnTGF- β 2 x *Il22*^{-/-} (Tg x *Il22*^{-/-}) CD4⁺ T cells. Tumor score and tumor number is shown. **C)** Representative endoscopic view and **D)** histological sections are shown; arrows indicate tumors; scale bar represents 500 μm. Results are cumulative from two independent experiments. Each dot represents one mouse (WT n= 18; Tg n= 14; *Il22*^{-/-} n= 19; Tg x *Il22*^{-/-} n= 7).

Lines indicate mean \pm sem; Tukey's multiple comparisons test was performed ($P < 0.05$) to assess the significance. **Source data are provided as a Source data file.**

2. For the AOM/DSS CRC mouse model, the authors transferred congenically marked wild type or TGF- β -DNR transgenic (Tg) CD4⁺ T cells into Rag1^{-/-} mice. This is concerning because of the lymphopenic niche in Rag1^{-/-} mice will lead to homeostatic proliferation of the transferred cells. It is important to determine if the same results would be achieved if the cells were transferred into mice with fully intact adaptive immune systems.

We thank the reviewer for this comment. Indeed, we confirmed the data using IL-17A^{Cre} x TGFBR1^{fl/fl} mice (Figure 4 B + C and Reply Figure 2, below) avoiding the transfer.

The reviewer is right that the lymphopenic niche in Rag1^{-/-} mice will lead to homeostatic proliferation of the transferred cells. Nevertheless, we intentionally used this approach since the TGF- β -DNR transgenic mice have impaired TGF- β signaling, not only on CD4⁺ T cells but also on CD8⁺ T cells, which are also important in tumor immunity. Unfortunately, we cannot do the transfer directly into immune competent mice: TGF- β -DNR transgenic mice were generated using the human CD2 promoter. Thus, these cells will be rejected if transferred into immune competent mice. We actually tried this experiment, and we could not monitor the cells for more than two weeks after transfer.

Reply Figure 2: TGF- β signaling in Th17 cells promotes IL-22 production.

Colitis associated colon cancer was induced in IL-17A^{Cre} x TGFBR2^{fl/fl} or IL-17A^{Cre} x TGFBR2^{wt/wt} mice. Production of IL-17A and IL-22 by T cells from colon was analyzed by flow cytometry. Each dot represents one mouse (TGFBR2^{Wt/Wt} n=9; TGFBR2^{Flox/Flox} n=7). Lines indicate mean +/- sem; Mann-Whitney test was performed to assess significance.

3. In Figure 5 in addition to using Ahr inhibitor, the data need to be substantiated by using AhR-deficient CD4+ T cells.

We agree with the reviewer. We thus imported Rorgt^{Cre} x AhR^{fl/fl} x ROSA^{YFP} mice (Kiss, Vonarbourg et al. 2011), which have a deletion of AhR in all T cells (Eberl and Littman, 2004), and performed the requested experiment. Of note, we could confirm our data using these mice, shown in Figure 5 C (also below).

'In order to confirm our results, we differentiated naïve T cells from Rorgt^{Cre} x AhR^{fl/fl} x ROSA^{YFP} mice [58], which have a deletion of AhR in all T cells [59]. The deletion of AhR in T cells significantly reduced the emergence of IL-17+IL-22- and IL-17+IL-22+ cells whereas IL-17-IL-22+ were not affected.' (page: 9, line: 308)

Figure 5: AhR signaling mediated the effects of TGF- β 1 on the emergence of IL-17+IL-22+ T cells.

A) Naïve T cells were differentiated in the presence of anti-CD3 (3 μ g/ml), APCs and indicated factors. Relative *Ahr*, *Rorc*, *cmaf*, *Il22*, *Il17a*, *Il10* mRNA expression on day 2 of the culture was measured using RT-PCR. 2-way ANOVA, Tukey's multiple comparisons test. **B)** Naïve T cells from *Foxp3^{mRFP} x IL-17A^{eGFP} x IL-22^{sgBFP}* reporter mice were cultured under Th17 polarizing condition (mAb IL-4 (10 μ g/ml), mAb INF- γ (10 μ g/ml), mAb CD3 (3 μ g/ml), mAb CD28 (0.5 μ g/ml), IL-6 (10 ng/ml), TGF- β 1 (1 ng/ml) and FICZ (100mM)) with increasing amounts of AhR antagonist (as indicated) for 4 days. Frequency of indicated cell populations are shown. Bars represent mean, error bars show +/- sem. One-way ANOVA, Dunnett's multiple comparisons test. Data are cumulative from three independent experiments. **C)** Naïve T cells from *Rorg1^{Cre} x AhR^{fl/fl} x ROSA^{YFP}* and control mice were cultured under Th17 polarizing condition for 4 days. Frequency of indicated cell populations are shown. Bars represent mean, error

bars show +/- sem. 2-way ANOVA, Sidak's multiple comparisons test. Data are cumulative from three independent experiments. **D)** *In vitro* differentiated Th17 cells (IL-17A+IL-22-) from Foxp3^{mRFP} x IL-17A^{eGFP} x IL-22^{sgBFP} reporter mice were sorted and re-cultured with a standard stimulation (mAb CD3 (3 µg/ml), mAb CD28 (0.5 µg/ml), IL-6 (10 ng/ml)) in the presence or absence of TGF-β and Ahr ligand (FICZ) or antagonist. Representative FACS plots upon re-culture in indicated conditions (left panel) and statistics (right panel) are shown. 2-way Anova, Bonferroni's multiple comparisons test. Bars represent mean, error bars show +/- sem. Data are cumulative from at least four independent experiments. **Source data are provided as a Source data file.**

Other comments:

1. From the methods, MACS kits were used for cell isolations but no post-enrichment purity was shown.

We did use MACS isolation kits in order to enrich for naïve T cells. Overall the purity was about 85% as shown in the representative Reply Figure 4 bellow. We have included this information in the revised version of the manuscript:

'To that end, we differentiated naïve T cells, purified with an efficiency around 85% using MACS beads, from wild type mice under different conditions, that have been previously reported to modulate IL-22 production in vitro' (page: 5, line: 187)

Reply Fig. 4: Enrichment of CD4+ naïve T cells using Magnetic cell separation (MACS) kits.

Mouse splenocytes were depleted from CD25 and CD44 and followed by positive CD4 selection. Frequency of enriched naïve CD4 + T cells is shown.

Moreover, we have also performed differentiation experiments using FACS-sorted naïve T cells, with a purity of around 95 - 98% in order to corroborate that our results do not depend on the isolation method used to enrich the naïve T cells (Reply Figure 5).

Reply Figure 5: TGF- β 1 promotes the differentiation from FACS-sorted naïve T cells of IL-17+IL-22+ producing T cells *in vitro*.

CD3⁺CD4⁺CD44⁻CD62L⁺ splenocytes from Foxp3^{mRFP} x IL-17A^{eGFP} x IL-22^{sgBFP} reporter mice were sorted and differentiated in the presence of mAb IL-4 (10 μ g/ml), mAb INF- γ (10 μ g/ml), mAb CD3 (3 μ g/ml), mAb CD28 (0.5 μ g/ml), IL-6 (10 ng/ml), APCs and indicated factors. Frequency of living CD4⁺ T cells producing IL-17A and IL-22 are shown after 5 days of culture.

2. In general, FACS data on IL-22 expression in transfer models need improvement. It needs special attention for *Citrobacter rodentium* model in which IL-22 can barely be detected.

We agree with the reviewer and we improved the FACS data shown in the manuscript including the one from the *Citrobacter rodentium* model (Figure S5).

REVIEWERS' COMMENTS:

Reviewer #1 (Remarks to the Author):

The revised manuscript by Perez et al reported a novel mechanism by which TGF- β promotes IL-22 production from Th17 cells and therefore augments the development of colitis associated colorectal cancer. In addition to the work described in their first submission, the authors now also show that: 1) TGF- β promotes the production of IL-22 in Th17 cells; 2) IL-22 from Th17 cells are important for colonic tumor development; and 3) TGF- β signaling to differentiated Th17 cells is required for the maintenance of AhR expression. These additional data were backed by compelling in vivo and in vitro evidence, and supports the overarching conclusion of the paper. The authors have addressed all questions of this reviewer, and the work is now in a great shape for publication.

Reviewer #2 (Remarks to the Author):

In the initial submission, three concerns were raised by this reviewer regarding the type of cancer (colitis-associated versus sporadic), inconsistent data from figure to figure, and potential expression of Foxp3 in IL-17(+) IL-22(+) T cells. These concerns have been satisfactorily addressed in this revised manuscript by additional data.

Reviewer #3 (Remarks to the Author):

The authors have eloquently addressed all reviewer comments; therefore, I believe this manuscript is worthy of acceptance to Nature Communications.

The causal roles of IL-17+IL-22+ vs IL-17-IL-22+ in the development of CRC have been clarified by performing further experiments which provide evidence to support their claims. By transferring WT vs IL22-/- and TGF- β -DNR transgenic x IL22-/- CD4+ T cell into an IL22-/- environment the authors demonstrate more tumor load in the mice that received WT but similar tumor load (less) in all other groups suggesting the effect is indeed IL-22 dependent. Additionally, the authors accept the limitation of their study concerning the contribution of IL-17A and have revised the manuscript accordingly.

As requested, by including an Ahr knockout control, the authors confirmed their results showing Ahr signaling mediated the effects of TGF- β 1 on the development of IL-17+IL-22+ T cells.

Minor comments were also addressed well.

Post-enrichment purity is now shown and the FACS data showing IL-22 expression was improved.

Point by point reply

Reviewer #1:

The revised manuscript by Perez et al reported a novel mechanism by which TGF- β promotes IL-22 production from Th17 cells and therefore augments the development of colitis associated colorectal cancer. In addition to the work described in their first submission, the authors now also show that: 1) TGF- β promotes the production of IL-22 in Th17 cells; 2) IL-22 from Th17 cells are important for colonic tumor development; and 3) TGF- β signaling to differentiated Th17 cells is required for the maintenance of AhR expression. These additional data were backed by compelling in vivo and in vitro evidence, and supports the overarching conclusion of the paper. The authors have addressed all questions of this reviewer, and the work is now in a great shape for publication.

We want to thank the reviewer for her/his positive assessment of our work

Reviewer #2:

In the initial submission, three concerns were raised by this reviewer regarding the type of cancer (colitis-associated versus sporadic), inconsistent data from figure to figure, and potential expression of Foxp3 in IL-17(+) IL-22(+) T cells. These concerns have been satisfactorily addressed in this revised manuscript by additional data.

We thank this reviewer by acknowledging the relevance of our new data.

Reviewer #3:

The authors have eloquently addressed all reviewer comments; therefore, I believe this manuscript is worthy of acceptance to Nature Communications.

The causal roles of IL-17+IL-22+ vs IL-17-IL-22+ in the development of CRC have been clarified by performing further experiments which provide evidence to support their claims. By transferring WT vs Il22^{-/-} and TGF- β -DNR transgenic x Il22^{-/-} CD4⁺ T cell into an Il22^{-/-} environment the authors demonstrate more tumor load in the mice that received WT but similar tumor load (less) in all other groups suggesting the effect is indeed IL-22 dependent. Additionally, the authors accept the limitation of their study concerning the contribution of IL-17A and have revised the manuscript accordingly.

As requested, by including an Ahr knockout control, the authors confirmed their results showing Ahr signaling mediated the effects of TGF- β 1 on the development of IL-17+IL-22+ T cells.

Minor comments were also addressed well.

Post-enrichment purity is now shown and the FACS data showing IL-22 expression was improved.

We are grateful for the overall positive assessment of our manuscript and for the opportunity to submit a revised version. The reviewer's comments have helped to further improve our study, and we are glad that they feel that our work is now acceptable for publication in *Nature Communications*.